# Learning from Synthetic Data Improves Multi-hop Reasoning

**Anmol Kabra**[1,*] **Yilun Yin**[1]**, Albert Gong**[1]**, Kamilė Stankevičiūtė**[1,2]**, Dongyoung Go**[1]**,
Johann Lee**[1]**, Katie Z. Luo**[3]**, Carla P. Gomes**[1]**, Kilian Q. Weinberger**[1]
[1]Cornell University [2]University of Cambridge [3]Stanford University

## Abstract

Reinforcement Learning (RL) has been shown to significantly boost reasoning capabilities of large language models (LLMs) in math, coding, and multi-hop reasoning tasks. However, RL fine-tuning requires abundant high-quality verifiable data, often sourced from human annotations, generated from frontier LLMs, or scored by LLM-based verifiers. All three have considerable limitations: human-annotated datasets are small and expensive to curate, LLM-generated data is hallucination-prone and costly, and LLM-based verifiers are inaccurate and slow. In this work, we investigate a cheaper alternative: RL fine-tuning on *rule-generated synthetic data* for multi-hop reasoning tasks. We discover that LLMs fine-tuned on synthetic data perform significantly better on popular real-world question-answering benchmarks, despite the synthetic data containing only fictional knowledge. On stratifying performance by question difficulty, we find that synthetic data teaches LLMs to *compose knowledge*—a fundamental and generalizable reasoning skill. Our work highlights rule-generated synthetic reasoning data as a free and scalable resource to improve LLM reasoning capabilities.

## 1 Introduction

Reinforcement learning (RL) has emerged as a powerful approach for enhancing reasoning capabilities of large language models (LLMs), with notable successes in math, coding, and logical reasoning (Bai et al., 2022; Lambert et al., 2025; Guo et al., 2025a; Guan et al., 2025). Central to this progress is the paradigm of RL with verifiable rewards (RLVR; Guo et al., 2025a; Lambert et al., 2025), where models' candidate answers are scored against known ground-truth answers or graded with verification tools. As a consequence, RLVR has been immensely effective in domains with abundant verifiable data or access to rule- or LLM-based verifiers. Yet RLVR faces a fundamental data bottleneck: high-quality training data with verifiable answers is scarce and expensive to curate, and LLM verifiers are slow and inaccurate (Lambert et al., 2025; Gong et al., 2025). As LLM training outpaces the availability of high-quality human-written text (Villalobos et al., 2024; Muennighoff et al., 2023), the community has responded with synthetic data (Liu et al., 2024)—either distilled from stronger models (Abdin et al., 2025) or generated as augmented reasoning traces (Trinh et al., 2024; Ruan et al., 2025). Both approaches, however, still depend on LLMs and inherit their verification challenges, high inference cost, and potential for contamination with pretraining corpora (Su et al., 2025).

We pursue a more radical alternative: ***rule-generated synthetic data***: questions generated from templates, context-free grammars, and logic programs on entirely fictional entities. Unlike LLM-generated synthetic data, rule-generated synthetic data is semantically simple by design, yet fully verifiable by construction. It can be produced at arbitrary scale, on any standard computer, for free.

The gap between such synthetic data and real-world counterparts is stark. A synthetic question from PhantomWiki (Gong et al., 2025) asks *"Who is the nephew of the friend of the person whose hobby is birdwatching?"*, while a question from HotpotQA (Yang et al., 2018) demands *"Aside from Yodobashi, what other towns were merged into the ward which gave the major Japanese electronics retail chain its name?"*. The former contains no real facts and uses templates, while the latter

---

*Correspondence: `anmol@cs.cornell.edu`. Code: `github.com/kilian-group/phantom-reasoning`.

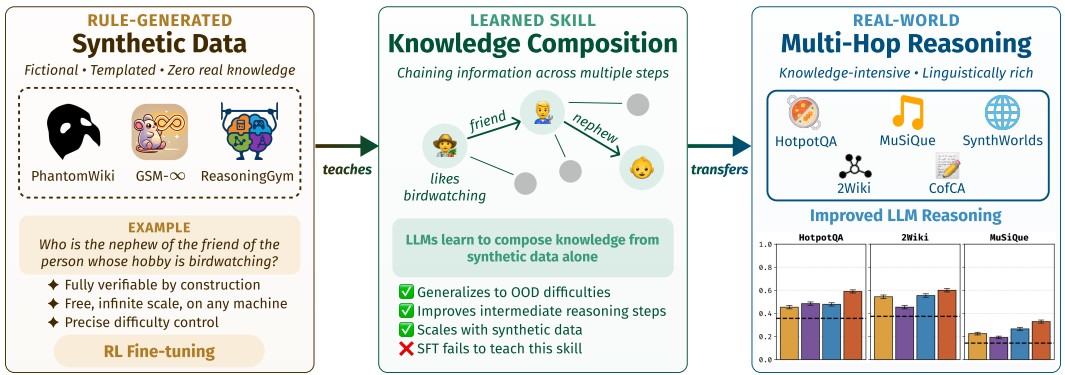

Figure 1: **Synthetic-to-real transfer by learning knowledge composition.** We RL fine-tune LLMs on rule-generated synthetic datasets (left). Despite containing no real-world knowledge, training on these datasets teaches models to compose knowledge (center). This fundamental skill of chaining information across multiple steps transfers to real-world multi-hop reasoning benchmarks (right).

requires navigating real-world knowledge and rich linguistic structure. It is *not obvious* whether rule-generated synthetic data can teach skills useful for the real-world (Stojanovski et al., 2025; Liu et al., 2024). Hence, we investigate the fundamental research question: ***can models develop general reasoning capabilities solely from synthetic data, without relying on real-world knowledge or linguistic complexity?*** Evidence for such a synthetic-to-real transfer would have transformative implications: *unlocking rule-generated synthetic data as a viable resource for scaling LLM reasoning*.

We present strong evidence for synthetic-to-real transfer in multi-hop reasoning. Among various reasoning capabilities, we focus on ***knowledge composition***: the elementary skill of integrating information across multiple steps. Through extensive RL fine-tuning experiments with rule-generated synthetic datasets (Gong et al., 2025; Zhou et al., 2025; Stojanovski et al., 2025), we demonstrate consistent transfer of multi-hop reasoning capabilities to real-world benchmarks (Yang et al., 2018; Ho et al., 2020; Trivedi et al., 2022; Wu et al., 2025a; Gu et al., 2025). The transfer is remarkable: a Qwen3-0.6B model RL fine-tuned on PhantomWiki (Gong et al., 2025) achieves relative F1 score improvements of 56% to 131% (Figure 2). This is despite the synthetic training data containing no factual knowledge useful to the real-world benchmarks. Performance transfer trends are consistent across synthetic data generators, real-world multi-hop benchmarks, and model families and sizes.

We further investigate *why* this transfer works. Recent work suggests that RL fine-tuning could be reinforcing and surfacing memorized pretraining knowledge (Shao et al., 2025; Yue et al., 2025). However, the fictional worlds in rule-generated synthetic data share zero knowledge overlap with reality, and models cannot improve by composing pretraining knowledge. Instead, they must learn to *compose knowledge* as a domain-independent skill. We confirm this by testing on held-out fictional worlds and out-of-domain (OOD) question complexities never seen during fine-tuning (Figure 3). Models extrapolate beyond training knowledge and question complexity levels, suggesting that they bootstrap this skill and generalize, rather than merely memorizing solution patterns. We find that Supervised fine-tuning (SFT) on rule-generated synthetic solution traces improves synthetic task performance but fails to generalize to the real-world (Table 2): it overfits to the training task rather than teaching a fundamental reasoning skill.

We also observe that this skill concretely manifests in model generations, illustrating *how* the transfer works. As RL fine-tuning progresses on synthetic data, models generate reasoning traces containing higher instances of correct intermediate answers (Figure 4). This points to emergent grounded generations in the real-world arising from outcome-only supervision with synthetic data. Finally, performance on real-world benchmarks scales monotonically with the number of synthetic training samples, with no sign of overfitting (Figure 5). This posits rule-generated synthetic data as a viable resource for scaling LLM reasoning capabilities with RL (Liu et al., 2024).

Taken together, our work spotlights rule-generated synthetic data as a free and scalable resource for teaching LLMs knowledge composition—a fundamental skill for real-world multi-hop reasoning.

## 2 RELATED WORK

**Reasoning in LLMs.** While LLM reasoning is a long-standing research area, the definition and assessment of reasoning capabilities is ambiguous and complex (Xie et al., 2024; Han et al., 2025). Thinking and reasoning models like DeepSeekMath (Shao et al., 2024), DeepSeek-R1 (Guo et al., 2025a), or Phi-4-mini-reasoning (Abdin et al., 2025) are typically evaluated in their reasoning skills through performance on various benchmarks. Current benchmarks span technical domains such as mathematics, coding, and puzzle-solving (Hendrycks et al., 2021; MAA; Cobbe et al., 2021; Jain et al., 2025; Chollet et al., 2025), knowledge-intensive tasks in science and law (Rein et al., 2024; Sawada et al., 2023), commonsense, abductive, and counterfactual reasoning (Talmor et al., 2019; Zhao et al., 2023; Bhagavatula et al., 2020; Wu et al., 2025b; Hüyük et al., 2025), multi-hop question-answering (Trivedi et al., 2022; Yang et al., 2018; Ho et al., 2020; Tang & Yang, 2024; Qi et al., 2021; Gu et al., 2025), and environment-based planning and tool use (Patil et al., 2024; Zhuang et al., 2023; Yao et al., 2024). Many of these benchmarks require decomposing the question into intermediate subproblems and composing their answers (Gong et al., 2025; Xie et al., 2025). This knowledge composition skill is considered to be an intrinsic property of effective reasoning models (Gandhi et al., 2025).

**Training and fine-tuning LLMs.** LLM performance on reasoning benchmarks can be improved with training or fine-tuning using several classes of techniques. The simplest approach is to train on the datasets directly using supervised fine-tuning (SFT; Lambert et al., 2025) with next-token prediction objective. This includes extensions to add more helpful instructions or to encourage a more detailed thinking process, for instance through instruction fine-tuning (Chung et al., 2024) and chain-of-thought (CoT) modeling (Xiang et al., 2025; Zelikman et al., 2022; Hao et al., 2025; Yao et al., 2023; Chen et al., 2023; Wan et al., 2025). Reinforcement learning from human feedback (RLHF; Christiano et al., 2017; Ouyang et al., 2022) offers a more powerful alternative, with algorithms including policy gradient-based PPO (Schulman et al., 2017), and variants such as GRPO (Shao et al., 2024) and DPO (Rafailov et al., 2023), among others (Hu et al., 2025; Pang et al., 2024; Brantley et al., 2025; Yu et al., 2025; Liu et al., 2025b; Shrivastava et al., 2025). When ground-truth answers are available, the reward model can be replaced by a fast verification function, an approach called Reinforcement learning with verifiable rewards (RLVR; Lambert et al., 2025). RLVR has been adopted by several recent reasoning models (Lambert et al., 2025; Guo et al., 2025a; Abdin et al., 2025), though its efficacy for eliciting novel reasoning in LLMs remain an open question (Wen et al., 2025; Yue et al., 2025; Shao et al., 2025; Zhao et al., 2025b).

**Leveraging synthetic data.** Real-world reasoning benchmarks conflate multiple skills (e.g., multi-hop reasoning with arithmetic reasoning or coding), making it difficult to isolate specific LLM capabilities (Wu et al., 2025c; Xie et al., 2024; Yu et al., 2024). Moreover, as LLMs are trained at internet scale, existing benchmarks become increasingly prone to test-set leakage (Gong et al., 2025; Wu et al., 2025c;b). Synthetic datasets address both issues: they can isolate targeted reasoning skills while providing unlimited fresh examples with verifiable rewards (Allen-Zhu, 2025; Zhao et al., 2025a; Liu et al., 2024; Yang et al., 2025; Sprague et al., 2025).

For training LLMs, recent work leverages LLM-generated synthetic data to augment human-curated data (Yang et al., 2025; Goldie et al., 2025; Huang et al., 2025; Saad-Falcon et al., 2024; Li et al., 2025). However, LLM-generated data is expensive, inherits verification challenges, and risks contamination with pretraining knowledge.

Rule-generated synthetic datasets—generated using templates, context-free grammars, and logic programs—have conventionally been used to evaluate reasoning capabilities in math (Mirzadeh et al., 2025; Zhou et al., 2025; Wu et al., 2025c), logic puzzles (Xie et al., 2024; Shojaee et al., 2025; Stojanovski et al., 2025), and natural language question-answering (Gong et al., 2025; Guo et al., 2025b; Sinha et al., 2019). It is fully verifiable by construction and free to produce at arbitrary scale. Yet whether training on such semantically simple data can teach skills useful for knowledge-intensive real-world reasoning remains underexplored (Yu et al., 2024; Mizrahi et al., 2025; Abbe et al., 2024b;a; Stojanovski et al., 2025). This is the question we study in this work.

## 3 METHODOLOGY

To study reasoning capabilities acquired from synthetic data, we RL fine-tune four LLMs: Qwen3-0.6B, Qwen3-1.7B, Qwen2.5-1.5B-Instruct (Qwen Team, 2024; 2025), and Phi-4-mini-reasoning (Abdin et al., 2025). By covering a range of model families and parameter sizes (0.6 to 4 billion parameters), we control for differences in pre-training and post-training strategies. This helps us isolate the effect of rule-generated synthetic data as the training signal for RLVR.

### 3.1 SYNTHETIC TRAINING DATASETS

Recent works in RLVR highlight the need for large datasets with two important characteristics: *scalable verification of model generations*, and *questions of varying difficulty* (Guo et al., 2025a; Wen et al., 2025; Shao et al., 2025; Lambert et al., 2025; Abdin et al., 2025). Scalable verification is essential for on-policy RL algorithms, because the reward function needs to be evaluated on-the-fly. Further, a mix of easy and hard questions is equally important: easy questions help the algorithm discover rewards early, while hard questions expand the frontier of reasoning capability. With these criteria in mind, we focus on rule-generated synthetic data.

**Rule-generated synthetic data.** Rule-generated synthetic data relies on strict templates, context-free grammars, and logic programs. This makes it semantically simple yet fully verifiable by construction, unlike LLM-generated data that has rich linguistic style but lacks verifiability. Because rule-generated synthetic data requires no GPU or API access, it can be produced at arbitrary scale, for free, and on any standard computer. This is valuable for designing LLM training pipelines under resource constraints. Generator programs also enable precise control over question complexities, which is crucial for RLVR and fine-grained reasoning analysis that we conduct in Section 4.

We use four rule-generated synthetic datasets that together cover diverse reasoning styles: **PhantomWiki** (synthetic multi-hop, Gong et al., 2025), **GSM-∞** (math, Zhou et al., 2025) (math), ReasoningGym's **Family-Relationships**, and **Knights-Knaves** (logical, Stojanovski et al., 2025). All datasets use templates on randomly-generated fictional worlds and contain no factual overlap with real-world multi-hop reasoning benchmarks. This allows us to study how reasoning styles of different kinds transfer to knowledge-intensive multi-hop reasoning, without any influence from shared factual knowledge. We discuss their individual features and generation settings below.

**PhantomWiki** (Gong et al., 2025) is a framework for generating datasets of natural language document corpora and question-answer pairs. Each PhantomWiki dataset represents a random universe of fictional people. Their personal attributes and inter-personal relations are described in Wikipedia-like documents. PhantomWiki uses a context-free grammar and logic programming-based algorithm to generate multi-hop reasoning questions with verifiable answers. Questions in PhantomWiki may have multiple answers unlike the other three datasets; they also require greater retrieval and knowledge composition skills. For example, answering "*Who is the nephew of the friend of the person who likes birdwatching?*" requires identifying all people who like birdwatching, and nephews of each of their friends. With PhantomWiki, we investigate the importance of training on synthetic data that align with the target real-world task, in our case: multi-hop question-answering.

We configure PhantomWiki datasets to only contain immediate family and friend relations, so that the "hops" are conceptually simple. We further filter out aggregation questions of the form "*How many ...*", to constrain the datasets to purely multi-hop questions like "*Who is the <relation> of ...?*" and "*What is the <attribute> of ...?*". This setup ensures that answering a question of difficulty $d$ requires hopping through exactly $d$ documents, and eliminates the confounding counting skill. To generate questions with varying difficulties, we generate 34 random universes each with 25 individuals, and set the context-free grammar recursion depth to 20. This process yields 330 questions per universe with question difficulties ranging from 1 to 9. We select 31 universes containing 10K samples for training, and reserve 3 universes of $\approx$ 1K samples for testing.

**GSM-Infinite** (GSM-∞; Zhou et al., 2025) generalizes the GSM8K benchmark—a collection of grade school math word problems (Cobbe et al., 2021)—to infinitely many questions. GSM-∞ builds a random computation graph on demand to represent the ground-truth solution trace. It then converts the graph to a word problem via natural language templates, which mimic common themes in GSM8K. With GSM-∞, we study transfer from arithmetic math-based reasoning to knowledge-intensive multi-hop reasoning.

We generate math word problems with 2–20 arithmetic operations (binary, no distractor facts), yielding $\approx$600 questions per difficulty level. Of the $\approx$12.5K total problems, we use a random subset of 10K samples for training and reserve the rest for testing.

**ReasoningGym** (Stojanovski et al., 2025) is an open-source RLVR library spanning domains including algebra, logic, and combinatorial games. We generate from two environments that probe complementary forms of logical reasoning. **RG-Family** (Family-Relationships) requires inferring the relationship between two individuals in a randomly generated family tree—the inverse of PhantomWiki, which infers individuals given a relationship. **RG-Knights** (Knights-Knaves) requires identifying truth-tellers from liars given a set of logical statements from randomly generated individuals. Together, these two datasets let us study whether logical deductive reasoning can teach skills relevant to multi-hop reasoning. We generate 10K training samples from each environment: RG-Family from family graphs of sizes uniformly sampled between 3 and 20, and RG-Knights balanced evenly across configurations of 2–6 people.

## 3.2 RL Fine-tuning for Reasoning

We use the Group Relative Policy Optimization (GRPO; Shao et al., 2024) RL fine-tuning algorithm. GRPO has been introduced as a variant of proximal policy optimization (PPO; Schulman et al., 2017). Where the PPO algorithm estimates the advantage term in its objective using a value model, the GRPO algorithm uses a group of completions for each prompt. We use the `GRPOTrainer` implementation from the open-source Hugging Face TRL v0.21.0 library (von Werra et al., 2020), which implements a special case of GRPO: the advantage is calculated per batch on each GPU, and KL-divergence penalty hyperparameter $\beta$ is set to 0. See Section A.1 for further details.

## 3.3 Prompt and Reward Design

We fine-tune LLMs to perform in-context reasoning, i.e. to answer questions given *all* the relevant context in the prompt. The prompt first includes the *evidence*: for a GSM-$\infty$ or a RG-Family or a RG-Knights question, this is the problem statement; for a PhantomWiki question, the evidence is the set of all 25 articles in the randomly generated PhantomWiki universe. After the evidence, our prompt includes an instruction for the LLM to output the final answer within `<answer>...</answer>` tags, which have been popularly used by DeepSeek-R1 (Guo et al., 2025a) and the Qwen3 family (Qwen Team, 2025). To further ground the answer output format, we append chain-of-thought (CoT) examples. For GSM-$\infty$, we use 3 automatically-generated ground-truth CoT from the training set; for PhantomWiki we use the 11 CoT examples curated by Gong et al. (2025); for RG-Family and RG-Knights we write 11 CoT examples ourselves. Finally, we pose the question to the LLM (our full prompts are included in Appendix C).

We extract the model's prediction from the final `<answer>...</answer>` tags, and compare it with ground-truth. For GSM-$\infty$, RG-Family, and RG-Knights questions, we assign a binary reward based on exact match. PhantomWiki questions can have multiple answers, so we assign rewards based on the F1 score of predictions. We randomly shuffle the 10K training samples in each dataset.

## 3.4 Evaluation Benchmarks

We evaluate on 5 real-world multi-hop reasoning datasets. For all these datasets we use the distractor versions, where the supporting information includes non-relevant (distracting) paragraphs. This evaluation setup measures LLMs' in-context reasoning as all questions can be answered given the context. We randomly subsample 500 question-answers from the respective test sets for evaluation.

1. **HotpotQA** (Yang et al., 2018) is a multi-hop question answering dataset containing over 100K questions. Each question follows a consistent 2-hop reasoning structure and requires combining information typically from 2 Wikipedia paragraphs.
2. **2WikiMultihopQA** (Ho et al., 2020) is a recent 2-hop dataset, containing over 190K questions organized into four categories: compositional, inference, comparison, and bridge-comparison. Like HotpotQA, the questions are grounded in Wikidata's knowledge graph, and follow a specific 2-hop path between related entities.
3. **MuSiQue** (Trivedi et al., 2022) evaluates compositional reasoning with 2-4 hop questions created by bridging single-hop questions, each using information from a supporting para-

graph. We use the MuSiQue-Answerable split of the dataset to ensure that all questions can be answered using a subset of the given context.

4. **CounterfactualQA (CofCA)** (Wu et al., 2025a) is a new rewritten subset of 2-4 hop questions from HotpotQA, 2WikiMultihopQA and MuSiQue. Wu et al. (2025b) manually rewrite questions to remove factual knowledge from the Wikipedia-based datasets, which LLMs could have memorized from to shortcut multi-hop reasoning.

5. **SynthWorlds-RM** (Gu et al., 2025) is a higher-complexity dataset of 2-6 hops and constraints curated from the Wikidata knowledge graph using graph motifs.

## 4 RESULTS

**Performance transfer from synthetic to real-world datasets.** Rule-generated synthetic datasets use strict yet simple templates on fictional worlds (Section 3.1), and differ drastically in grammar, semantics, and linguistic complexity from knowledge-intensive multi-hop reasoning in the real-world (Section 3.4). Synthetic-to-real transfer is therefore not obvious (Stojanovski et al., 2025). Yet, as Figure 2 shows, RL fine-tuning on rule-generated synthetic data remarkably improves performance on *all* real-world evaluation benchmarks. This transfer is consistent across model families and sizes, with the smallest models benefiting the most. Notably, Qwen3 (Qwen Team, 2025, Sec. 4) and Phi-4-mini-reasoning (Abdin et al., 2025, Sec. 3,4) already underwent extensive training on synthetic reasoning data during post-training, yet they still improve with continued RL fine-tuning on rule-generated synthetic data. In other words, ***learning to compose knowledge in fictional worlds devoid of linguistic diversity transfers to realistic multi-hop tasks rich in semantic complexity***. We note that RL fine-tuning on in-domain real-world data still outperforms synthetic-to-real transfer (Figure 7), though such data is expensive to curate and limited in scale.

**RL fine-tuning augments performance beyond answer formatting.** Improved performance from RL fine-tuning could partly stem from learning to write answers in the right format, rather than from improved underlying reasoning (Shao et al., 2025). We disentangle these two effects with an ablation study. We RL fine-tune all small models for 3K steps with binary reward for using `<answer>...</answer>`. Table 1 shows that this format-only training does not help Qwen3 or Phi-4-mini-reasoning, but substantially improves Qwen2.5-1.5B-Instruct.

There are two takeaways from this ablation. First, RL fine-tuning can teach answer formatting. Qwen2.5-1.5B-Instruct learns "reward hacking" with format-only reward. Therefore, the Qwen2.5-1.5B-Instruct's synthetic-to-real transfer in Figure 2 is due to both learning output formatting and answering correctly. Second, and more importantly, RL fine-tuning teaches knowledge composition. Qwen3 and Phi-4-mini-reasoning already produce correct output formatting at initialization, so format-only training does not improve performance. Therefore, *all* of their synthetic-to-real transfer in Figure 2 must come from learning to compose knowledge. This confirms that ***LLMs can develop knowledge composition from synthetic data alone***.

|  |  | HotpotQA | 2WikiMultihopQA | MuSiQue |
|---|---|---|---|---|
| Qwen3-0.6B | base | $0.36 \pm 0.02$ | $0.37 \pm 0.02$ | $0.14 \pm 0.01$ |
|  | format | $0.38 \pm 0.02$ | $0.34 \pm 0.02$ | $0.13 \pm 0.01$ |
| Qwen3-1.7B | base | $0.59 \pm 0.02$ | $0.64 \pm 0.02$ | $0.34 \pm 0.02$ |
|  | format | $0.64 \pm 0.02$ | $0.67 \pm 0.02$ | $0.35 \pm 0.02$ |
| Phi-4-mini-reasoning | base | $0.48 \pm 0.02$ | $0.66 \pm 0.02$ | $0.27 \pm 0.02$ |
|  | format | $0.47 \pm 0.02$ | $0.48 \pm 0.02$ | $0.26 \pm 0.02$ |
| Qwen2.5-1.5B-Instruct | base | $0.02 \pm 0.01$ | $0.14 \pm 0.02$ | $0.04 \pm 0.01$ |
|  | format | $0.43 \pm 0.02$ | $0.30 \pm 0.02$ | $0.20 \pm 0.02$ |

Table 1: **Ablation study on fine-tuning with binary format reward.** F1 scores of Qwen3 and Phi-4-mini-reasoning LLMs do not improve when RL fine-tuned with binary reward for using `<answer>...</answer>`. Qwen2.5-1.5B-Instruct improves remarkably with binary format reward. We report standard error over the evaluation datasets.

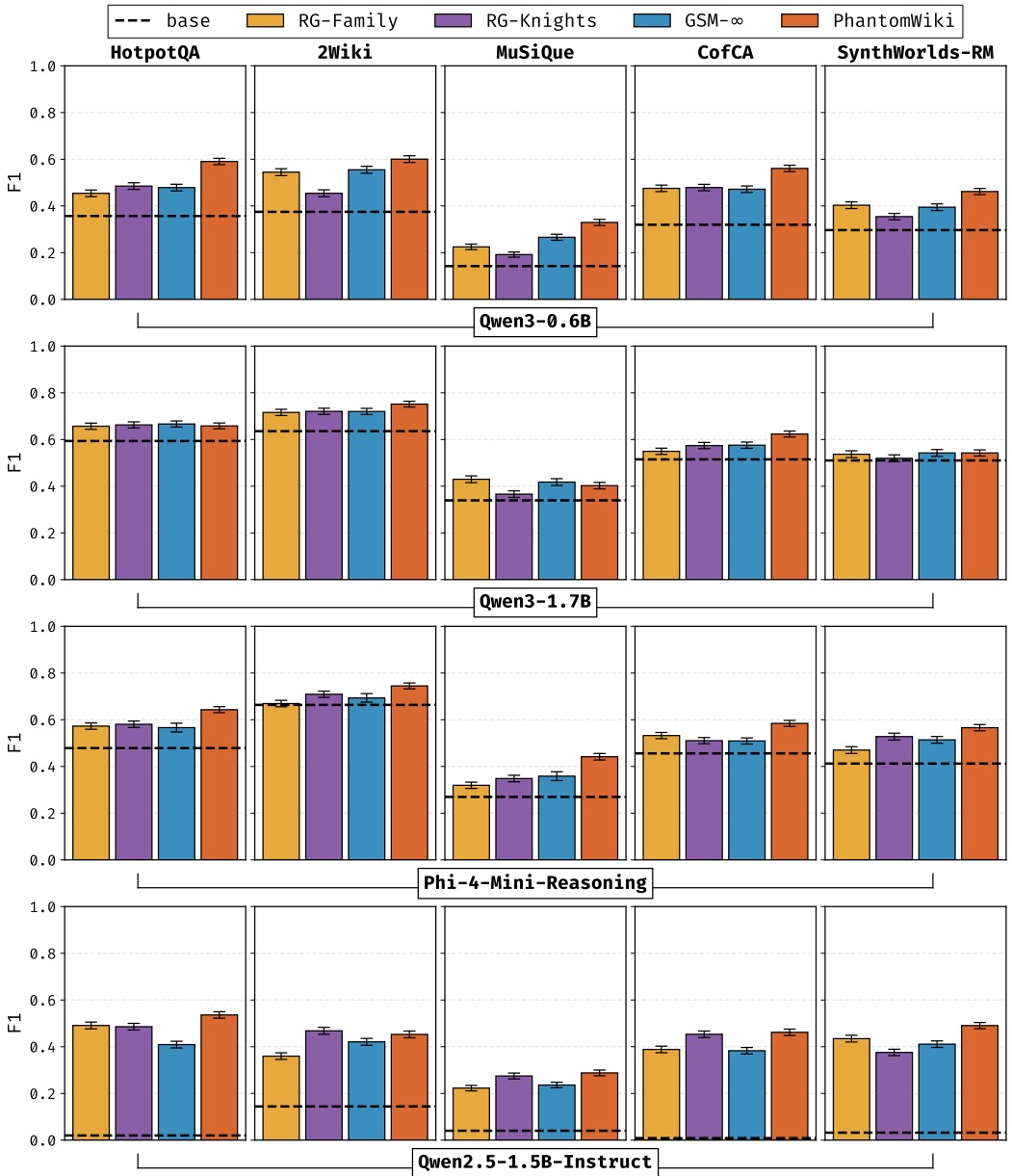

Figure 2: **F1 scores on real-world multi-hop reasoning datasets of LLMs RL fine-tuned on synthetic datasets.** We observe consistent transfer from synthetic data training to real-world multi-hop reasoning benchmarks: {RG-Family, RG-Knights, GSM-∞, PhantomWiki} → {HotpotQA, 2WikiMultihopQA, MuSiQue, CofCA, SynthWorlds-RM}. The performance transfer trends hold across model families and sizes (Qwen and Phi LLMs in 0.6-4B parameter range). On each synthetic dataset, we fine-tune each LLM with 2 random training seeds, and evaluate final checkpoints of both experiment runs. With this we calculate the standard error, shown as error bars. See Figure 6 for similar plots for larger LLMs Qwen3-4B and Qwen2.5-7B-Instruct.

**Knowledge composition generalizes to unseen knowledge and question difficulties.** The real-world multi-hop reasoning benchmarks are derived from Wikipedia, which was likely part of each model's pretraining corpus. It is possible that the synthetic-to-real transfer in Figure 2 comes from better composition of memorized knowledge, rather than from learning a general composition skill. Indeed, Shao et al. (2025); Yue et al. (2025) hypothesize that RL fine-tuning primarily surfaces memorized pretrained knowledge in LLMs. To test whether models acquire knowledge composition as

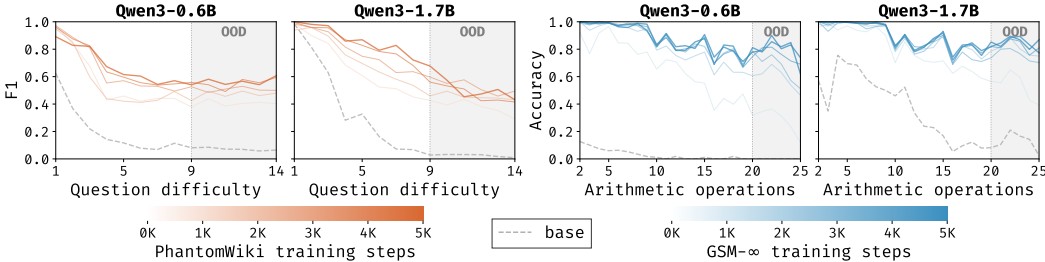

Figure 3: **Performance of intermediate training checkpoints on unseen test sets, stratified by question complexity.** We evaluate Qwen3-0.6B and Qwen3-1.7B intermediate checkpoints trained on PhantomWiki (left) and GSM-∞ (right) on held-out test sets. Test sets share no factual overlap with training data. Question complexity is defined as the number of document hops (PhantomWiki) or arithmetic operations (GSM-∞) required to reach the answer. As fine-tuning progresses (lines growing darker), performance improves across all complexity levels, including out-of-domain (OOD) difficulties beyond those seen during training. See Figure 8 for similar plots for other LLMs.

a *fundamental* skill, we evaluate on held-out PhantomWiki and GSM-∞ test sets. These share zero factual overlap with the corresponding training datasets and include question complexities beyond those seen during RL fine-tuning (Gong et al., 2025; Zhou et al., 2025).

Figure 3 stratifies performance of Qwen3-0.6B and Qwen3-1.7B intermediate training checkpoints by question complexity. Since the factual knowledge in held-out PhantomWiki and GSM-∞ test sets are fully disjoint from training data, models cannot succeed by surfacing parametric knowledge. They must learn to compose knowledge as a ***domain-independent and fundamental skill***. Moreover, models improve across all difficulty levels as fine-tuning progresses—including out-of-domain (OOD) complexities never encountered during fine-tuning. This OOD generalization highlights that models can ***bootstrap the skill beyond what they were trained on***.

**SFT fails to teach knowledge composition.** Supervised fine-tuning (SFT) requires reasoning traces, either from human annotations, frontier LLMs, or ground-truth solution traces. The first two are expensive to collect, while ground-truth solution traces are rarely available in rule-generated synthetic data, which typically only generates questions and verifiable answers. This makes RL fine-tuning more broadly applicable than SFT for leveraging rule-generated synthetic data.

Even so, when ground-truth solution traces *are* available, can SFT deliver synthetic-to-real transfer like RL fine-tuning? To answer this, we SFT Qwen3-0.6B and Qwen3-1.7B models on GSM-∞ training data, which includes solution traces (Zhou et al., 2025). We evaluate the fine-tuned models on held-out GSM-∞ test set and real-world HotpotQA benchmark, and compare SFT with RL fine-tuning in Table 2. While SFT improves performance on synthetic GSM-∞ data, we observe no performance improvement in real-world multi-hop reasoning. On the other hand, RL fine-tuning yields both in-domain and transfer improvement. ***SFT can thus overfit to the synthetic data task and fail to teach knowledge composition necessary for transfer***.

| | Accuracy on GSM-∞ | | | F1 score on HotpotQA | | |
|---|---|---|---|---|---|---|
| | base | SFT | RL | base | SFT | RL |
| Qwen3-0.6B | 0.0241 | **0.7735** | 0.6452 | 0.3569 | 0.3995 | **0.4786** |
| Qwen3-1.7B | 0.1354 | 0.7742 | **0.8532** | 0.5935 | 0.5761 | **0.6664** |

Table 2: **Performance of SFT vs RL fine-tuning on synthetic (GSM-∞) and real-world (HotpotQA) test sets.** SFT on ground-truth solution traces from the GSM-∞ training data improves performance on the synthetic task but not on the target HotpotQA. RL fine-tuning improves models on both tasks, indicating that it teaches transferable knowledge composition where SFT does not.

**Synthetic data fine-tuning leads to grounded generations in real-world reasoning.** Our fine-tuning setup only rewards the final answer, unlike process reward modeling that supervise intermediate steps (Lightman et al., 2023). Figure 3 shows that knowledge composition improves final-

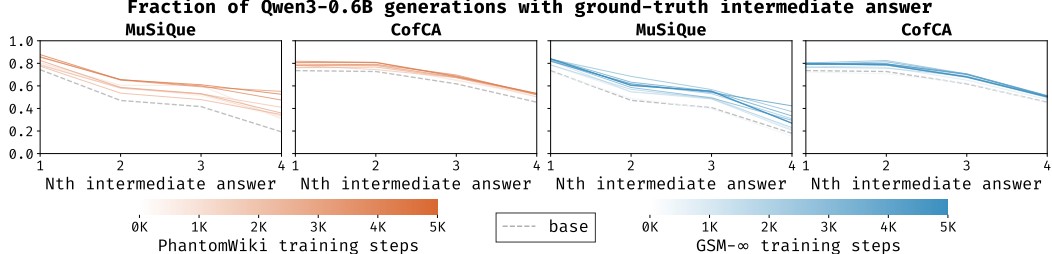

Figure 4: **Groundedness of Qwen3-0.6B reasoning traces in MuSiQue and CofCA intermediate answers.** For each dataset, we plot the fraction of model generations (reasoning traces) that contain the ground-truth $n^{th}$ intermediate answer. As RL fine-tuning progresses on PhantomWiki (left) and GSM-$\infty$ (right), reasoning traces include a higher proportion of correct intermediate answers. This indicates that RL fine-tuning with outcome-only reward on synthetic data also improves the reasoning process. See Figure 10 for similar analysis on other models.

answer accuracy across all question complexities; however, it does not reveal whether the *reasoning process* itself improves, i.e., whether intermediate steps become more accurate.

To investigate this, we leverage the fact that each MuSiQue and CofCA question comes with intermediate sub-questions and corresponding ground-truth answers. In Figure 4, we measure how often these intermediate answers appear in the model's reasoning traces across training checkpoints. As RL fine-tuning progresses on PhantomWiki and GSM-$\infty$, models produce reasoning traces with increasingly higher proportions of correct intermediate answers. That is, outcome-only reward on synthetic data actually improves the *groundedness* of the reasoning process—a useful byproduct. Together with Figures 2 and 3, this shows that **knowledge composition improves LLM generation quality in both final-answer correctness and intermediate-step groundedness**.

**Performance scales with synthetic data.** Recent work scales RL fine-tuning by modifying the RL algorithm, model architecture, and test-time compute (Liu et al., 2025a). We highlight an additional scaling axis: *training data*. Since we train for a single epoch on each synthetic dataset, models see each training sample exactly once. So, evaluating intermediate checkpoints is equivalent to studying **synthetic data scaling in RL fine-tuning**.

In Figure 5, Qwen3-0.6B steadily improves on all real-world benchmarks with more training steps, or equivalently, more synthetic training data. We do not observe overfitting to the synthetic task, in contrast to SFT (Table 2). We see a similar trend for other LLMs in Figure 9. Notably, different models exhibit varying *malleability* to RL fine-tuning: smaller models such as Qwen3-0.6B start weaker but improve steeply, while larger models such as Qwen3-1.7B are stronger at initialization but improve more gradually. Understanding how model initialization influences RL fine-tuning remains an open problem (Yue et al., 2025; Liu et al., 2024).

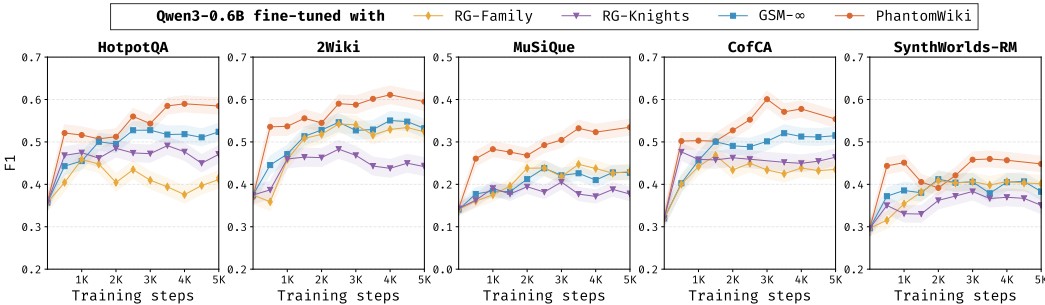

Figure 5: **Real-world multi-hop reasoning performance of Qwen3-0.6B intermediate training checkpoints.** We evaluate checkpoints every 500 training steps, and report mean $\pm$ standard error with the solid line and shaded region. Performance on all benchmarks improves steadily with training steps—equivalently, with the number of synthetic training samples—providing evidence for synthetic data scaling in RL fine-tuning.

## 5 DISCUSSION

**Transferability of Reasoning.** Our findings show that RL fine-tuning on fictional, template-based datasets improves performance on real-world multi-hop reasoning benchmarks. Because the synthetic training data shares zero factual overlap with evaluation domains, this cross-domain transfer rules out surface-level memorization and supports knowledge composition as a transferable skill (Toplak & Stanovich, 2002). More broadly, our results suggest that such multi-step logical inference is a domain-independent skill that can be learned in isolation from real-world knowledge.

However, the extent of this transferability remains an open question. Real-world reasoning tasks require both factual knowledge and compositional skill, and models can learn memorization and generalization simultaneously (Xie et al., 2024). While memorization can hurt performance in counterfactual settings (Wu et al., 2025b), it may complement composition in standard settings. Understanding how synthetic and knowledge-intensive real-world data interact during training is an important direction for future work.

**Synthetic datasets as meaningful training signals.** Our analysis demonstrates that rule-generated synthetic datasets are effective *training* signals: models learn transferable reasoning capabilities from them. This has practical implications. As high-quality human-annotated reasoning data becomes increasingly scarce (Villalobos et al., 2024; Muennighoff et al., 2023), synthetic datasets offer a scalable alternative where domain experts design verifiable task generators rather than labeling individual examples (Stojanovski et al., 2025; Liu et al., 2024; Yang et al., 2025).

While our work focuses on knowledge composition for multi-hop reasoning, synthetic data could teach other reasoning skills as well (Stojanovski et al., 2025; Allen-Zhu, 2025; Liu et al., 2024; Xiong et al., 2025; Sprague et al., 2025; Yuan et al., 2025). Future work could explore whether causal reasoning, counterfactual inference, or analogical thinking exhibit similar transferability. Identifying the boundary conditions for this transfer—when it works, when it fails, and how it interacts with other training objectives—remains an important open (Zhao et al., 2023; Wu et al., 2025c).

## 6 CONCLUSION

We investigate rule-generated synthetic datasets as a scalable alternative to real-world training data for LLM reasoning. Our results demonstrate that such synthetic training data teaches the transferable knowledge composition skill that yields significant gains on diverse real-world multi-hop benchmarks. This is despite sharing zero knowledge overlap with evaluation domains and using semantically simple language. We show that this transfer reflects genuine skill acquisition: models generalize to unseen question difficulties, produce more grounded reasoning traces, and continue improving with more synthetic data. Our findings highlight new avenues for scaling RL fine-tuning, overcoming traditional data bottlenecks.

### ACKNOWLEDGMENTS

AK thanks fellow Cornell CS students Atharv Sonwane, Chao Wan, Laura Zielinski, and Linxi Zhao for manuscript feedback and discussions on formal logic. AK thanks Claude for help with plotting. DG is supported by Empire AI Postdoctoral Fellowship. Authors acknowledge compute resources from the National Artificial Intelligence Research Resource (NAIRR) Pilot, Purdue Anvil AI, and NVIDIA's DGX Station compute platform through their Early Access Program. This work is supported by the National Science Foundation (NSF) grants RI-2530143, OAC-2118310, and through the AI Research Institutes program Award No. DMR-2433348. This work was partially supported by funding from NewYork-Presbyterian for the NYP-Cornell Cardiovascular AI Collaboration, the National Institute of Food and Agriculture (USDA/NIFA), the Air Force Office of Scientific Research (AFOSR), and a Schmidt AI2050 Senior Fellowship, a Schmidt Sciences program. We thank anonymous reviewers for their helpful feedback.

### ETHICS STATEMENT

Our work adheres to the ICLR Code of Ethics, and does not pose any societal, personal, or organizational risks.

## REPRODUCIBILITY STATEMENT

To ensure reproducibility, we use free and open-source software and publicly available LLMs. Our code is available at github.com/kilian-group/phantom-reasoning, where we include our full dataset preparation, model training code, and evaluation configuration (overview in Methodology section and Appendix A). We further report standard errors in our results to show significance, generate data with fixed random seeds, and set fixed training random seeds where possible.

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

## A  EXPERIMENT CONFIGURATION

### A.1  GRPO DETAILS

We restate the GRPO objective from Shao et al. (2024) here. Given a question $q$ sampled from a distribution over question set $P(Q)$, GRPO samples a group of $G$ output completions $\{o_1, \ldots, o_G\}$ from the old LLM $\pi$ with parameters $\theta_{\text{old}}$. Then it assigns each output completion a scalar reward value $\{R_1, \ldots, R_G\}$. The algorithm estimates the advantage $\widehat{A}_i$ of each completion by normalizing with respect to the average reward as a baseline. The final objective is as follows:

$$\mathcal{J}_{\text{GRPO}}(\theta) = \mathbb{E}_{[q\sim P(Q),\{o_1,\ldots,o_G\}\sim\pi_{\theta_{\text{old}}}(\cdot|q)]}$$

$$\left[ \frac{1}{G}\sum_{i=1}^{G}\frac{1}{|o_i|}\sum_{t=1}^{|o_i|}\left\{\min\left[r_{i,t}\hat{A}_{i,t}, \text{clip}\left(r_{i,t}, 1-\varepsilon, 1+\varepsilon\right)\hat{A}_{i,t}\right] - \beta\mathbb{D}_{\text{KL}}[\pi_\theta||\pi_{\text{ref}}]\right\}\right]$$

where $\widehat{A}_{i,t} = \dfrac{R_i - \text{mean}(R_1,\ldots,R_G)}{\text{stdev}(R_1,\ldots,R_G)}$.

Here $\pi_{\text{ref}}$ is a reference policy (usually model initialization) used in the KL divergence penalty $\mathbb{D}_{\text{KL}}$, $\epsilon, \beta$ are hyperparameters, and the relative weight $r_{i,t}$ for output completion $o_i$ is calculated on a per-token basis $r_{i,t} = \frac{\pi_\theta(o_{i,t}|q,o_{i,<t})}{\pi_{\theta_{\text{old}}}(o_{i,t}|q,o_{i,<t})}$.

**Implementation.** We use the GRPOTrainer implementation from the open-source Hugging Face TRL v0.21.0 library (von Werra et al., 2020). We use vLLM colocate mode (Kwon et al., 2023) and FlashAttention-2 (Dao, 2024) on 4 NVIDIA H100 GPUs each with 80GB VRAM. With the configuration in Listing 1, RL finetuning a 1 to 4B parameter LLM on 10K training samples takes ≈1 day on our Linux cluster. [1] The prompt length varies for each training dataset, and we adjust the max_prompt_length to prevent prompt truncation:

1. PhantomWiki: 6000
2. GSM-∞: 2048
3. HotpotQA: 6000
4. 2WikiMultihopQA: 6000
5. MuSiQue: 8000

```
# Training parameters
per_device_train_batch_size: 8
gradient_accumulation_steps: 1
num_generations: 16

# vLLM settings
use_vllm: true
vllm_mode: "colocate"
vllm_gpu_memory_utilization: 0.20

# Generation parameters
max_completion_length: 4096
temperature: 1.0
top_p: 1.0
```

[1]The larger models Qwen3-1.7B and Phi-4-mini-reasoning take the full 1 day, i.e. using ≈100 H100 hours per training experiment as they generate long CoT. The Qwen2.5-1.5B-Instruct model does not generate long CoT, and thus trains in ≈20 H100 hours.

```
15 top_k: null
16 min_p: null
17 repetition_penalty: 1.0
18
19 # GRPO algorithm parameters
20 beta: 0.0
21 epsilon: 0.2
22 importance_sampling_level: "token"
23 scale_rewards: true
24 loss_type: bnpo
25 mask_truncated_completions: false
```

Listing 1: GRPOTrainer hyperparameter values in our YAML configuration file

**Adjusting hyperparameters for large LLMs.** Due to memory constraints on our GPU resources, for Phi-4-mini-reasoning and Qwen3-4B LLMs with 4 billion parameters each, we lower hyperparameter values `vllm_gpu_memory_utilization: 0.25`, `per_device_train_batch_size: 4`, and `num_generations: 8`. For Qwen2.5-7B-Instruct, we further reduce `vllm_gpu_memory_utilization: 0.30`, `per_device_train_batch_size: 2`, and `num_generations: 4`. When developing GRPO fine-tuning experiments on the small models, we had observed that a large number of GRPO rollouts and training batch size improved RL fine-tuning stability and performance. Unfortunately our resource constraints compel us to use suboptimal lower values of these hyperparameters. This could be factor in the mixed results on large LLMs in Figure 6. We leave it to future work to study the effect of rollouts and batch size on RL fine-tuning performance.

## A.2 SFT DETAILS

For SFT on ground-truth solution traces in the GSM-$\infty$ training data, we use the `SFTTrainer` implementation from the same open-source Hugging Face TRL v0.21.0 library (von Werra et al., 2020). Maximum length of 8192 is sufficient to accommodate each prompt and solution trace, as it exceeds `max_prompt_length + max_completion_length` equalling 6244 in Section A.1. The loss is computed on all text both prompt and the solution trace; computing loss on solution trace only did not materially change final performance. We use default values for other hyperparameters. With the configuration in Listing 2, SFT uses 1 H100 GPU with 80 GB VRAM.

```
1 # Training parameters
2 per_device_train_batch_size: 8
3 gradient_accumulation_steps: 1
4 packing: false
5 max_length: 8192
```

Listing 2: SFTTrainer hyperparameter values in our YAML configuration file

# B ADDITIONAL RESULTS

See Figures 6 to 10.

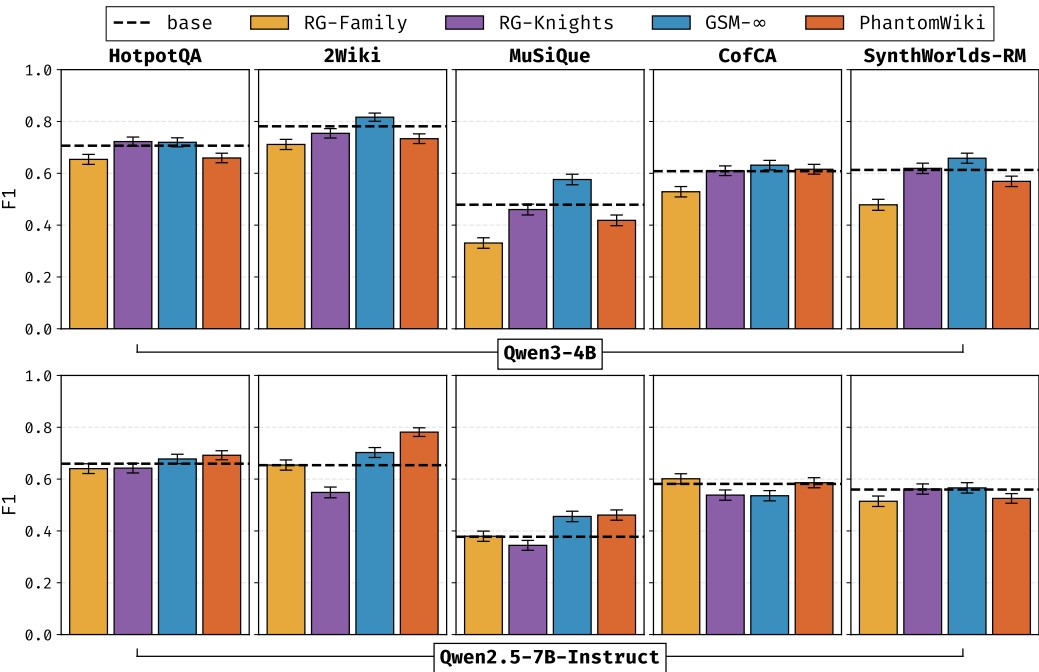

Figure 6: **F1 scores on real-world multi-hop reasoning datasets of LLMs RL fine-tuned on synthetic datasets.** Larger LLMs Qwen3-4B and Qwen2.5-7B-Instruct, which already perform strongly at initialization, show less consistent synthetic-to-real transfer. We note that memory constraints necessitated suboptimal hyperparameters for these models (Section A.1), and scaling up RL fine-tuning for larger models remains an important direction. On each synthetic dataset, we fine-tune each LLM with a fixed training seed and evaluate the final checkpoint. We report the standard error of evaluating the final experiment checkpoints.

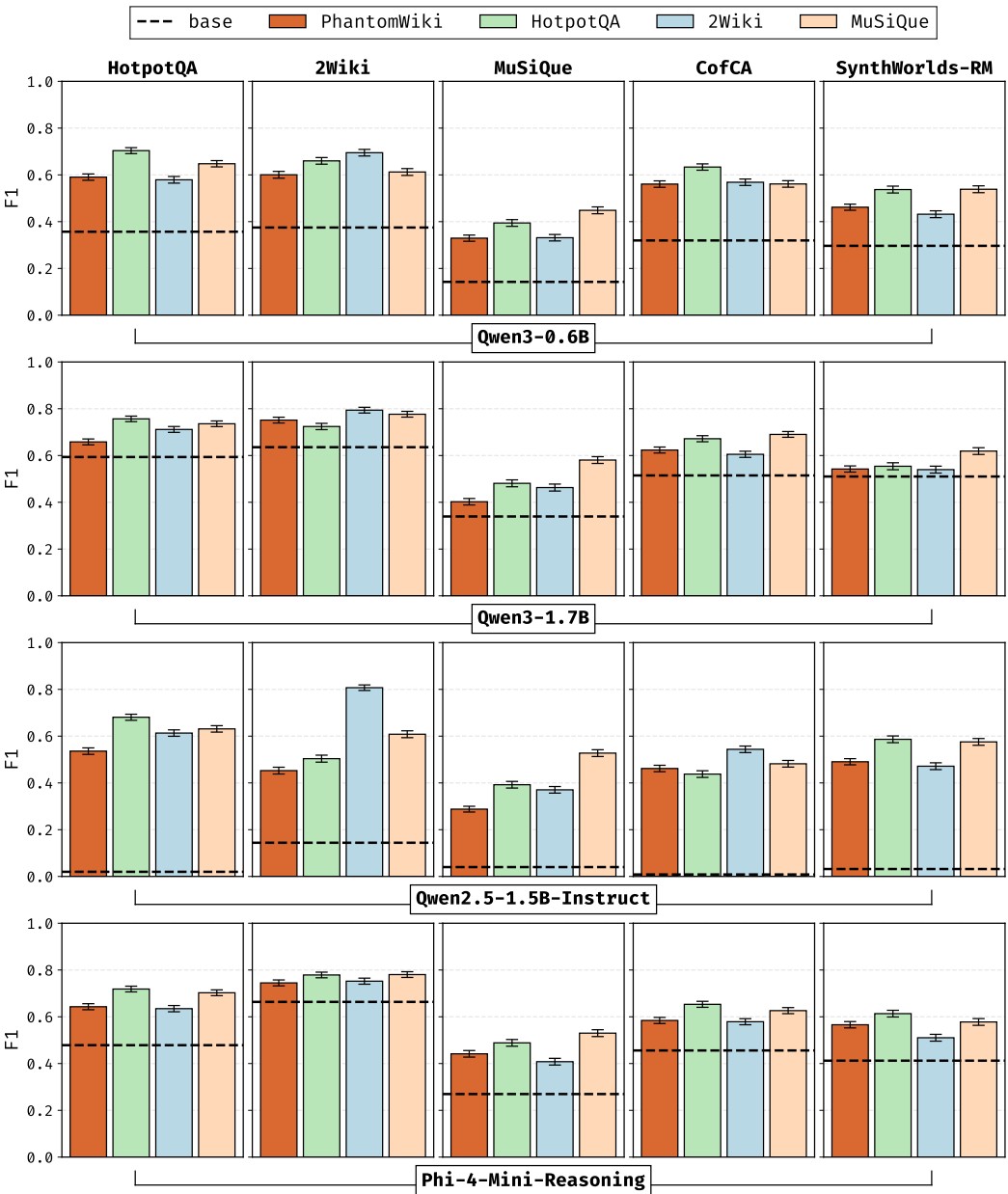

Figure 7: **F1 scores on real-world multi-hop reasoning datasets of LLMs RL fine-tuned on training sets of real-world data.** On each training dataset, we RL fine-tune each LLM with 2 random training seeds, following the same training setup as with synthetic datasets in Figure 2 (10K training samples, F1 scores as reward). RL fine-tuning on real-world data outperforms rule-generated synthetic data like PhantomWiki, which is expected as synthetic data lacks rich linguistic complexity of real-world data. In fact, in-domain training data outperforms other data sources, i.e., HotpotQA training data yields maximum gains for HotpotQA benchmark (light green beats others in the left most column), and similarly for 2WikiMultihopQA and MuSiQue. We report the standard error over both experiments and benchmark samples.

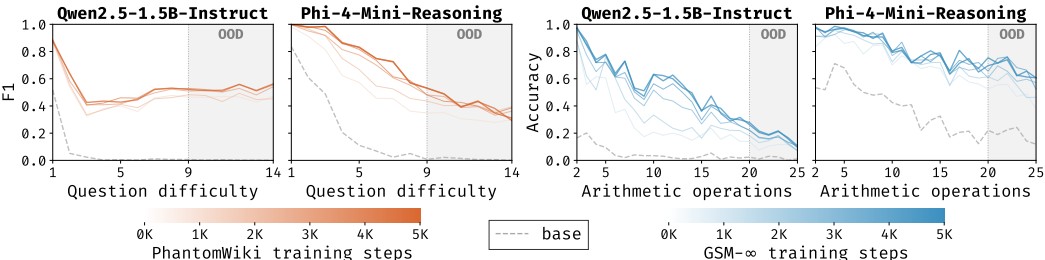

Figure 8: **Performance of intermediate training checkpoints on unseen test sets, stratified by question complexity.** We evaluate Qwen2.5-1.5B-Instruct and Phi-4-mini-reasoning intermediate checkpoints trained on PhantomWiki (left) and GSM-∞ (right) on held-out test sets. Test sets share no factual overlap with training data. Question complexity is defined as the number of document hops (PhantomWiki) or arithmetic operations (GSM-∞) required to reach the answer. As fine-tuning progresses (lines growing darker), performance improves across all complexity levels, including out-of-domain (OOD) difficulties beyond those seen during training.

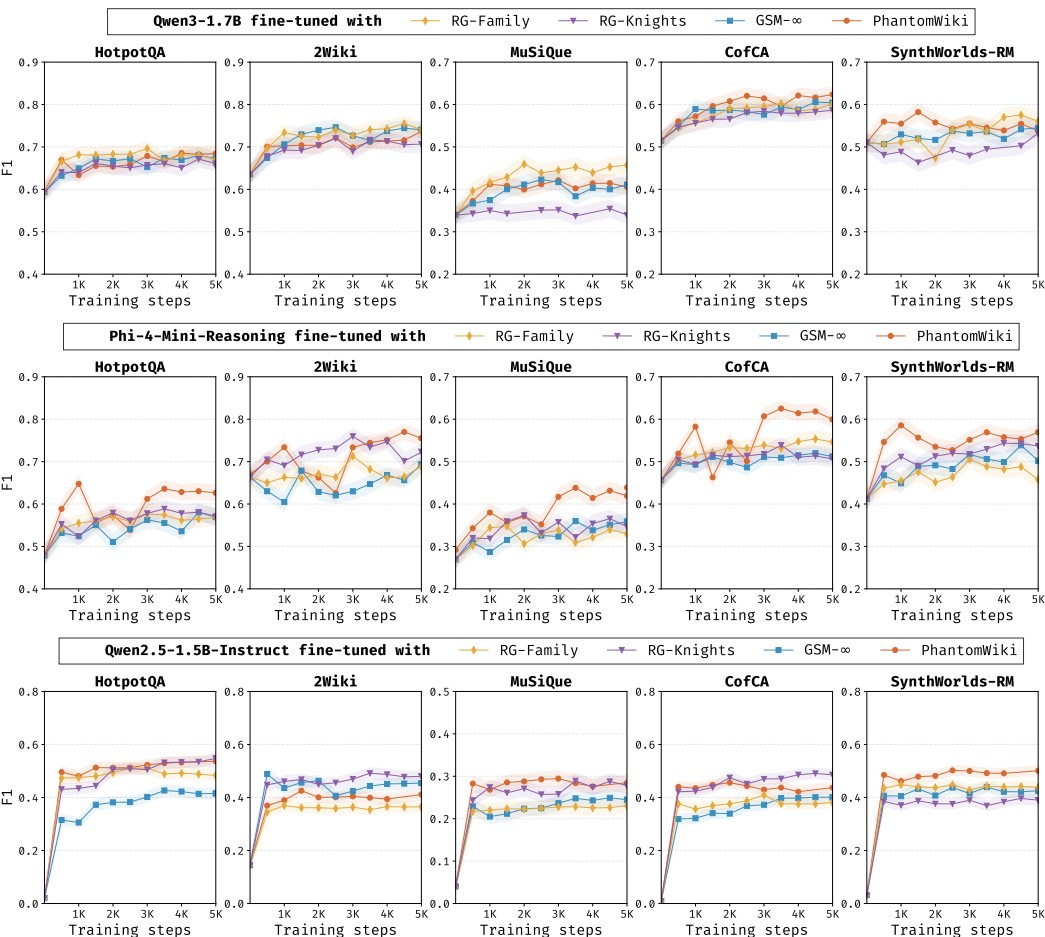

Figure 9: **Real-world multi-hop reasoning performance of LLM intermediate training checkpoints.** We evaluate checkpoints every 500 training steps, and report mean ± standard error with the solid line and shaded region. Performance of Qwen3-1.7B and Phi-4-mini-reasoning on all benchmarks improves steadily with training steps—equivalently, with the number of synthetic training samples—providing evidence for synthetic data scaling in RL fine-tuning. Qwen2.5-1.5B-Instruct performance saturates quickly.

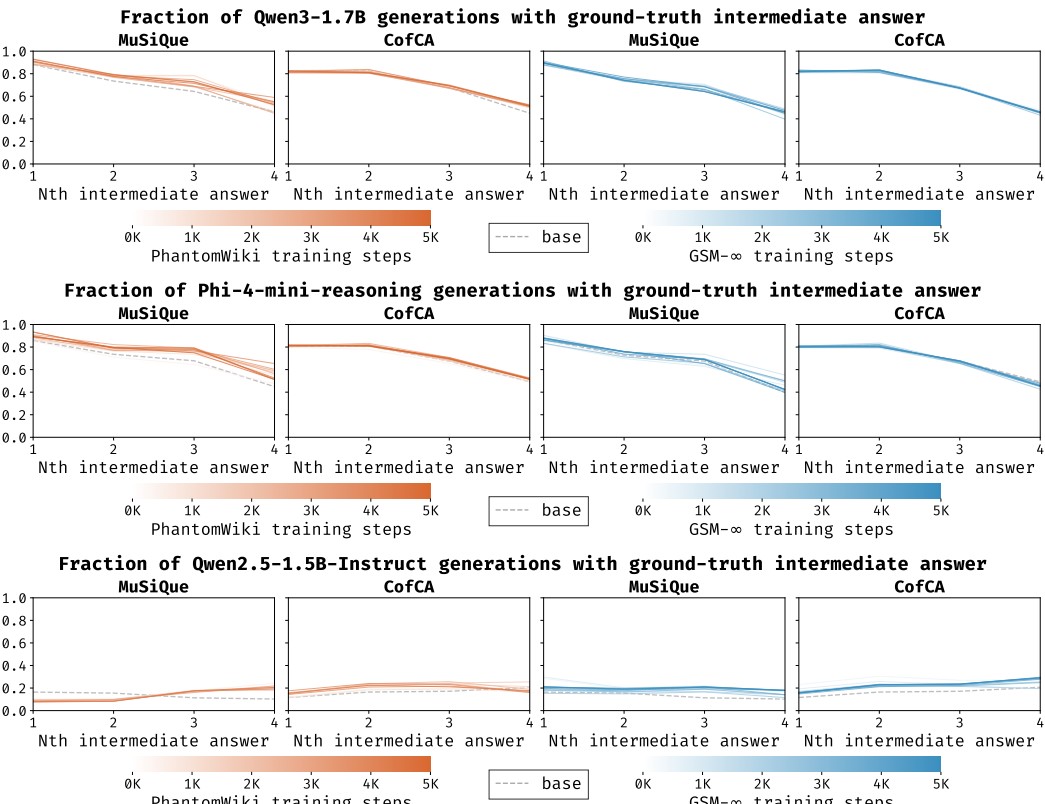

Figure 10: **Groundedness of LLM reasoning traces in MuSiQue and CofCA intermediate answers.** For each dataset, we plot the fraction of model generations (reasoning traces) that contain the ground-truth $n^{th}$ intermediate answer. As RL fine-tuning progresses on PhantomWiki (left) and GSM-$\infty$ (right), reasoning traces include a higher proportion of correct intermediate answers. This indicates that RL fine-tuning with outcome-only reward on synthetic data also improves the reasoning process. The effect is lower for larger LLMs on the CofCA benchmark.

## C  PROMPTS

### C.1  PHANTOMWIKI PROMPT

We use CoT prompt template and examples from ([Gong et al., 2025](#)), with a custom instruction asking the LLM to output the final answer within `<answer>...</answer>`.

```
You are given the following evidence:
(BEGIN EVIDENCE)
{{evidence}}
(END EVIDENCE)

You will be provided a question. Your response must end with the final answer enclosed in tags: <answer
    >FINAL_ANSWER</answer>

Here, FINAL_ANSWER must be one of the following:
- a name (if there is only one correct answer);
- a list of names separated by ',' (if there are multiple correct answers); or
- numbers separated by ',' (if the answer is numerical); or
- empty string (if there is no answer).

Here are some examples:
(START OF EXAMPLES)
Example 1:
Question: Who is the sister of Aida Wang?
Answer: Based on the evidence, the sisters of Aida Wang are Barabara Beltran, Vicki Hackworth. <answer>
    Barabara Beltran,Vicki Hackworth</answer>.

Example 2:
Question: Who is the child of Alvaro Smock?
Answer: Based on the evidence, the children of Alvaro Smock are Eli Smock, Gene Smock. <answer>Eli
    Smock,Gene Smock</answer>.

Example 3:
Question: Who is the friend of the child of Alvaro Smock?
Answer: First I need to find the children of Alvaro Smock. Based on the evidence, the children of
    Alvaro Smock are Eli Smock, Gene Smock. Now I need to find the friends of Eli Smock and Gene Smock
    . Based on the evidence, the friends of Eli Smock are Leisa Lutz, Shelli Beltran, Vicki Hackworth,
     Virgil Hackworth, Alison Smock, Brian Beltran. The friends of Gene Smock are Leeann Hackworth,
    Leisa Lutz, Ricardo Hackworth, Alvaro Smock, Dominique Smock. <answer>Leisa Lutz,Shelli Beltran,
    Vicki Hackworth,Virgil Hackworth,Alison Smock,Brian Beltran,Leeann Hackworth,Ricardo Hackworth,
    Dominique Smock</answer>.

Example 4:
Question: Who is the aunt of Vicki Hackworth?
Answer: An aunt is the sister of a parent. Based on the evidence, the parents of Vicki Hackworth are
    Shelli Beltran, Dino Beltran. To find the aunt of Vicki Hackworth, I need to find the sister of
    Shelli Beltran and Dino Beltran. Based on the evidence, Shelli Beltran has no sister, and the
    sister of Dino Beltran is Stacia Toombs. <answer>Stacia Toombs</answer>.

Example 5:
Question: What is the occupation of the husband of Stacia Toombs?
Answer: Based on the evidence, the husband of Stacia Toombs is Wilbert Toombs. The occupation of
    Wilbert Toombs is theatre manager. <answer>theatre manager</answer>.

Example 6:
Question: What is the hobby of the daughter-in-law of Lannie Smock?
Answer: A daughter-in-law is the wife of a child. Based on the evidence, the children of Lannie Smock
    are Eli Smock, Gene Smock. Eli Smock has no wife, and the wife of Gene Smock is Dominique Smock.
    The hobby of Dominique Smock is dominoes. <answer>dominoes</answer>.

Example 7:
Question: What is the date of birth of the person whose hobby is finance?
Answer: I need to search for people whose hobby is finance. Based on the evidence, the person whose
    hobby is finance is Stacia Toombs. The date of birth of Stacia Toombs is 0959-03-22. <answer
    >0959-03-22</answer>.

Example 8:
Question: Who is the great-granddaughter of the person whose occupation is biomedical scientist?
Answer: I need to search for people whose occupation is biomedical scientist. Based on the evidence,
    the person whose occupation is biomedical scientist is Lannie Smock. To find the great-
    granddaughter of Lannie Smock, I need to find the daughter of the child of the child of Lannie
    Smock. Based on the evidence, the children of Lannie Smock are Eli Smock, Gene Smock. Eli Smock
    has no child, and the child of Gene Smock is Williams Smock. The daughters of Williams Smock are
    Shelli Beltran, Stacia Toombs. <answer>Shelli Beltran,Stacia Toombs</answer>.

Example 9:
Question: How many friends does Ryan Wang have?
Answer: Based on the evidence, the friends of Ryan Wang are Shelli Beltran, Stacia Toombs, Virgil
    Hackworth, Aida Wang. <answer>4</answer>.

Example 10:
Question: How many friends does the child of Alvaro Smock have?
Answer: First, I need to find the children of Alvaro Smock. Based on the evidence, the children of
    Alvaro Smock are Eli Smock, Gene Smock. Now I need to find how many friends they have. Based on
    the evidence, the friends of Eli Smock are Leisa Lutz, Shelli Beltran, Vicki Hackworth, Virgil
    Hackworth, Alison Smock, Brian Beltran. The friends of Gene Smock are Leeann Hackworth, Leisa Lutz
    , Ricardo Hackworth, Alvaro Smock, Dominique Smock. <answer>6,5</answer>.
```

```
Example 11:
Question: How many uncles does the friend of Stacia Toombs have?
Answer: First, I need to find the friends of Stacia Toombs. Based on the evidence, the friends of
    Stacia Toombs are Brian Beltran, Isiah Lutz, Leeann Hackworth, Lesley Lutz, Ryan Wang. Now I need
     to find how many uncles they have. An uncle is the brother of a parent. Based on the evidence,
    Brian Beltran has no parents, Isiah Lutz has no parents, Leeann Hackworth has 2 parents, Lesley
    Lutz has 2 parents, and Ryan Wang has no parents. Based on the evidence, the parents of Leeann
    Hackworth are Vicki Hackworth, Ricardo Hackworth. But both parents do not have brothers. Based on
     the evidence, the parents of Lesley Lutz are Leisa Lutz, Isiah Lutz. The brother of Leisa Lutz is
     Virgil Hackworth, so he is an uncle of Lesley Lutz. Isiah Lutz has no brother. So the friends of
     Stacia Toombs have 0, 0, 0, 1, 0 uncles. Unique is 0, 1. <answer>0,1</answer>.
(END OF EXAMPLES)

Question: {{question}}
Answer: """
```

## C.2 GSM-∞ PROMPT

We modify the CoT prompt template from PhantomWiki (Gong et al., 2025) by replacing EVIDENCE with the problem statement. GSM-∞ also generates a templated solution for each question pairs, which we use as the CoT examples in the prompt.

```
You are given the following problem:
(BEGIN PROBLEM)
{{problem}}
(END PROBLEM)

You will be provided a question on the above problem. Your response must end with the final answer
    enclosed in tags: <answer>FINAL_ANSWER</answer>

Here, FINAL_ANSWER must be a number.

Here are some examples:
(START OF EXAMPLES)
Example 1:
Question: What is the total number of adult animals in Maple Creek?
Answer: Define adult wolf in Maple Creek as r; so r = 2. Define total number of adult animals in Maple
    Creek as p; so p = r = 2. <answer>2</answer>.

Example 2:
Question: What is the total number of schools in Clearwater Bay?
Answer: Define elementary school in Riverton City as b; so b = 3. Define private middle school in
    Clearwater Bay as i; so i = b = 3. Define public highschool in Clearwater Bay as M; so M = i = 3.
    Define elementary school in Clearwater Bay as G; so G = 2. Define total number of schools in
    Clearwater Bay as W; V = G + i = 2 + 3 = 5; so W = V + M = 5 + 3 = 8. <answer>8</answer>.

Example 3:
Question: What is the total number of movies in Festival de Clairmont?
Answer: Define upbeat metropolis comedy in Festival de Saint-Rivage as m; so m = 4. Define total number
     of movies in Festival de Saint-Rivage as k; so k = m = 4. Define intense detective thriller in
    Festival Lumi\u00e8re de Valmont as C; l = k - m = 4 - 4 = 0; so C = 3 + l = 3 + 0 = 3. Define
    total number of movies in Festival Lumi\u00e8re de Valmont as Q; so Q = C = 3. Define solemn
    period drama in R\u00eaves de Belleville as N; t = Q + C = 3 + 3 = 6; T = t + k = 6 + 4 = 10; so N
     = 4 + T = 4 + 10 = 14. Define total number of movies in R\u00eaves de Belleville as y; so y = N =
     14. Define futuristic sci-fi movie in Festival de Clairmont as A; z = y + N = 14 + 14 = 28; q = z
     + C = 28 + 3 = 31; so A = 3 * q = 3 * 31 = 93. Define total number of movies in Festival de
    Clairmont as p; so p = A = 93. <answer>93</answer>.
(END OF EXAMPLES)

Question: {{question}}
Answer:
```

## C.3 RG-FAMILY AND RG-KNIGHTS PROMPTS

We modify the CoT prompt template from PhantomWiki (Gong et al., 2025) by removing the EVIDENCE section, as below. We manually write several CoT examples from each synthetic training dataset.

```
You will be provided with a question. Your response must end with the final answer enclosed in tags: <
    answer>FINAL_ANSWER</answer>

Here are some examples:
(START OF EXAMPLES)
{{examples}}
(END OF EXAMPLES)

You are given the following problem:
Question: {{question}}
Answer:
```

### C.3.1 RG-FAMILY PROMPT EXAMPLES

```
Example 1:
Question: Harry is married to Emily. They have a child called Daniel.
What is Emily to Daniel? Respond only with the word that describes their relationship.
Answer: Harry and Emily are the parents of Daniel, so Emily is the mother of Daniel. <answer>mother</
      answer>

Example 2:
Question: James is married to Aria. They have a child called George. George is married to Olivia.
What is George to Olivia? Respond only with the word that describes their relationship.
Answer: Since George is married to Olivia, George is the husband of Olivia. <answer>husband</answer>

Example 3:
Question: John is married to Olivia. They have a child called Liam. Liam is married to Willow. They
      have a child called Logan.
What relation is John to Olivia? Answer with a single word.
Answer: John is married to Olivia, so John is the husband of Olivia. <answer>husband</answer>

Example 4:
Question: Ryder is married to Lily. They have a child called John. John is married to Aria. They have
      children called Noah and Daniel.
How is John related to Daniel? Provide the relationship in one word.
Answer: Noah and Daniel are children of John, so John is the father of Daniel. <answer>father</answer>

Example 5:
Question: Alexander is married to Lisa. They have a child called Joseph. Joseph is married to Luna.
      They have a child called Eleanor. William is married to Amelia. They have a child called Luna.
How is Eleanor related to Luna? Provide the relationship in one word.
Answer: Joseph is married to Luna, and together they have a child called Eleanor, so Eleanor is the
      daughter of Luna. <answer>daughter</answer>

Example 6:
Question: Charles is married to Eleanor. They have a child called Mason. Mason is married to Susan.
      They have children called Lucy and Ryder. Christopher is married to Patricia. They have a child
      called Susan.
What is Lucy to Mason? Respond only with the word that describes their relationship.
Answer: Lucy is the daughter of Mason, so Lucy is the daughter of Mason. <answer>daughter</answer>

Example 7:
Question: John is married to Barbara. They have children called Kai and Atlas. Atlas is married to
      River. Kai is married to Luna. They have a child called Joseph. Michael is married to Zoe. They
      have a child called Luna.
What is Luna to Kai? Respond only with the word that describes their relationship.
Answer: Luna is married to Kai, so Luna is the wife of Kai. <answer>wife</answer>

Example 8:
Question: Noah is married to Barbara. They have children called Aiden and Charles. Charles is married
      to Lisa. They have a child called River. Aiden is married to Lucy. They have a child called Atlas.
       Matthew is married to Sarah. They have a child called Lucy.
What relation is Noah to Atlas? Answer with a single word.
Answer: Noah has a child called Aiden. Moreover, Aiden is married to Lucy, who together have a child
      called Atlas. So Noah is the father of Aiden, and Aiden is the father of Atlas. Therefore, Noah is
       the grandfather of Atlas. <answer>grandfather</answer>

Example 9:
Question: Phoenix is married to Amelia. They have children called Lucas, Aiden and Sophia. Daniel is
      married to Willow. They have a child called Nova. Aiden is married to Grace. Lucas is married to
      Nova. They have a child called Sebastian. Sophia is married to Noah.
What relation is Sebastian to Lucas? Answer with a single word.
Answer: Lucas is married to Nova, and together they have a child called Sebastian. So Sebastian is the
      son of Lucas. <answer>son</answer>

Example 10:
Question: Sebastian is married to Ava. They have children called David, Aiden and Hannah. Thomas is
      married to Luna. They have a child called Karen. Aiden is married to Sky. David is married to
      Karen. They have a child called Matthew. Hannah is married to James. They have a child called
      Daniel.
What relation is Matthew to Karen? Answer with a single word.
Answer: David is married to Karen, and together they have a child called Matthew. So Matthew is the son
       of Karen. <answer>son</answer>

Example 11:
Question: Zion is married to Grace. They have children called Andrew, Logan and Patricia. Matthew is
      married to Emily. They have a child called Sophie. Logan is married to Nova. They have a child
      called Margaret. Andrew is married to Sophie. They have a child called Henry. Patricia is married
      to Lucas. They have a child called Karen.
How is Andrew related to Karen? Provide the relationship in one word.
Answer: Patricia has a child called Karen. Moreover, Andrew and Patricia are siblings, so Andrew is the
       uncle of Karen. <answer>uncle</answer>
```

## C.3.2   RG-KNIGHTS PROMPT EXAMPLES

```
Example 1:
Question: A very special island is inhabited only by heroes and villains. Heroes always tell the truth,
      and villains always lie. You meet 2 inhabitants: Benjamin, and Scarlett. Benjamin was heard
      saying, "if Benjamin is a hero then Scarlett is a hero". Scarlett stated, "Scarlett is a hero or
      Benjamin is a hero". So who is a hero and who is a villain? (Format your answer like: "Benjamin is
       a hero/villain, and Scarlett is a hero/villain")
Answer: Assume Benjamin is a hero (tells truth). His statement "if Benjamin is hero then Scarlett is
      hero" means Scarlett must be a hero. If Scarlett is a hero, her statement "Scarlett is hero OR
```

```
                Benjamin is hero" is true (both parts are true). This is consistent. <answer>Benjamin is a hero,
                and Scarlett is a hero.</answer>

Example 2:
Question: A very special island is inhabited only by altruists and egoists. Altruists always tell the
        truth, and egoists always lie. You meet 2 inhabitants: Luke, and Riley. In a statement by Luke: "
        if Riley is an egoist then Luke is an altruist". Riley remarked, "Luke is an egoist if and only if
         Riley is an altruist". So who is an altruist and who is an egoist? (Format your answer like: "
        Luke is a altruist/egoist, and Riley is a altruist/egoist")
Answer: Assume Luke is an egoist (lies). His statement "if Riley is egoist then Luke is altruist" is
        false. For a conditional to be false, the premise must be true and conclusion false. So Riley is
        an egoist and Luke is not an altruist (Luke is egoist), which is consistent. Now check Riley's
        statement "Luke is egoist <-> Riley is altruist". Since both Luke and Riley are egoists, left side
         is true and right side is false, making the biconditional false, which is consistent with Riley
        being an egoist who lies. <answer>Luke is an egoist, and Riley is an egoist.</answer>

Example 3:
Question: A very special island is inhabited only by angels and devils. Angels always tell the truth,
        and devils always lie. You meet 3 inhabitants: Logan, Aurora, and Riley. Logan asserted: "Aurora
        is an angel". As Aurora put it, "if Logan is an angel then Riley is a devil". Riley stated, "Logan
         is a devil". So who is an angel and who is a devil? (Format your answer like: "Logan is a angel/
        devil, Aurora is a angel/devil, and Riley is a angel/devil")
Answer: Assume Logan is an angel (tells truth). Then Aurora is an angel (as Logan stated). If Aurora is
        an angel, her statement "if Logan is angel then Riley is devil" means Riley is a devil. If Riley
        is a devil (lies), he says "Logan is a devil", which is false, consistent with lying. <answer>
        Logan is an angel, Aurora is an angel, and Riley is a devil.</answer>

Example 4:
Question: A very special island is inhabited only by heroes and villains. Heroes always tell the truth,
         and villains always lie. You meet 3 inhabitants: Luke, Henry, and Zoey. "Luke is a hero or Henry
        is a hero," Luke claimed. Henry said that if Zoey is a villain then Henry is a hero. In a
        statement by Zoey: "if Henry is a hero then Luke is a villain". So who is a hero and who is a
        villain? (Format your answer like: "Luke is a hero/villain, Henry is a hero/villain, and Zoey is a
         hero/villain")
Answer: Assume Luke is a hero (tells truth). His statement "Luke is hero OR Henry is hero" is true. Let
        's assume Henry is also a hero. Henry's statement "if Zoey is villain then Henry is hero" is true
        since Henry is a hero. Now for Zoey's statement "if Henry is hero then Luke is villain" - if Zoey
        is a villain (lies), her statement should be false. Since Henry is a hero and Luke is not a
        villain, the conditional is false, which is consistent with Zoey lying. <answer>Luke is a hero,
        Henry is a hero, and Zoey is a villain.</answer>

Example 5:
Question: A very special island is inhabited only by sages and fools. Sages always tell the truth, and
        fools always lie. You meet 4 inhabitants: Alexander, Elizabeth, Amelia, and Penelope. In a
        statement by Alexander: "if Amelia is a fool then Amelia is a sage". Elizabeth said that Penelope
        is a sage if and only if Amelia is a fool. "Alexander is a sage and Penelope is a sage" - Amelia.
        Penelope stated, "Amelia is a sage or Elizabeth is a fool". So who is a sage and who is a fool? (
        Format your answer like: "Alexander is a sage/fool, Elizabeth is a sage/fool, Amelia is a sage/
        fool, and Penelope is a sage/fool")
Answer: Alexander's statement "if Amelia is fool then Amelia is sage" is a contradiction if Amelia is a
        fool, so for Alexander to be a sage (truth-teller), Amelia must be a sage. If Amelia is a sage,
        her statement "Alexander is sage AND Penelope is sage" means both are sages. Penelope's statement
        "Amelia is sage OR Elizabeth is fool" is true if Penelope is a sage - since Amelia is sage, the OR
        is true. Elizabeth's statement "Penelope is sage <-> Amelia is fool" - if Penelope is sage and
        Amelia is sage (not fool), the biconditional is false, so Elizabeth is a fool. <answer>Alexander
        is a sage, Elizabeth is a fool, Amelia is a sage, and Penelope is a sage.</answer>

Example 6:
Question: A very special island is inhabited only by heroes and villains. Heroes always tell the truth,
         and villains always lie. You meet 4 inhabitants: Sophia, Alexander, Grace, and Liam. Sophia
        stated, "if Sophia is a hero then Alexander is a villain". "Grace is a villain if and only if Liam
         is a hero," Alexander mentioned. "Sophia is a villain if and only if Sophia is a hero," Grace
        declared. As Liam put it, "Grace is a villain and Liam is a hero". So who is a hero and who is a
        villain? (Format your answer like: "Sophia is a hero/villain, Alexander is a hero/villain, Grace
        is a hero/villain, and Liam is a hero/villain")
Answer: Grace's statement "Sophia is villain <-> Sophia is hero" is a contradiction, so it's false,
        meaning Grace is a villain. Assume Sophia is a hero. Her statement "if Sophia is hero then
        Alexander is villain" means Alexander is a villain. Alexander's statement "Grace is villain <->
        Liam is hero" - if Alexander is a villain (lies), this should be false. Since Grace is a villain,
        for the biconditional to be false, Liam must not be a hero (Liam is villain). Liam's statement "
        Grace is villain AND Liam is hero" - since Liam is a villain, this false statement is consistent.
        <answer>Sophia is a hero, Alexander is a villain, Grace is a villain, and Liam is a villain.</
        answer>

Example 7:
Question: A very special island is inhabited only by angels and devils. Angels always tell the truth,
        and devils always lie. You meet 5 inhabitants: Ava, Amelia, Daniel, Mia, and Jack. "Mia is an
        angel if and only if Jack is a devil," Ava mentioned. In Amelia's words: "Daniel is a devil or Mia
         is an angel". Daniel was heard saying, "Ava is a devil". Mia noted, "Jack is a devil". Jack was
        heard saying, "Mia is a devil if and only if Mia is an angel". So who is an angel and who is a
        devil? (Format your answer like: "Ava is a angel/devil, Amelia is a angel/devil, Daniel is a angel
        /devil, Mia is a angel/devil, and Jack is a angel/devil")
Answer: Jack's statement "Mia is devil <-> Mia is angel" is a contradiction, so it's false, meaning
        Jack is a devil. If Mia is an angel, her statement "Jack is devil" is true. Ava's statement "Mia
        is angel <-> Jack is devil" - both parts are true, so the biconditional is true, meaning Ava is an
         angel. Amelia's statement "Daniel is devil OR Mia is angel" - since Mia is angel, the OR is true,
         so Amelia is an angel. Daniel's statement "Ava is devil" is false since Ava is angel, so Daniel
        is a devil. <answer>Ava is an angel, Amelia is an angel, Daniel is a devil, Mia is an angel, and
        Jack is a devil.</answer>

Example 8:
```

```
Question: A very special island is inhabited only by saints and sinners. Saints always tell the truth,
    and sinners always lie. You meet 5 inhabitants: Penelope, Lily, Riley, Mia, and Aria. "Mia is a
    saint and Lily is a sinner," Penelope claimed. Lily noted, "Penelope is a sinner". Riley asserted:
    "if Lily is a sinner then Mia is a sinner". Mia noted, "Penelope is a sinner if and only if
    Penelope is a saint". Aria expressed that Lily is a saint. So who is a saint and who is a sinner?
    (Format your answer like: "Penelope is a saint/sinner, Lily is a saint/sinner, Riley is a saint/
    sinner, Mia is a saint/sinner, and Aria is a saint/sinner")
Answer: Mia's statement "Penelope is sinner <-> Penelope is saint" is a contradiction, so it's false,
    meaning Mia is a sinner. Assume Lily is a saint. Lily's statement "Penelope is sinner" means
    Penelope is a sinner. Penelope's statement "Mia is saint AND Lily is sinner" - since Mia is sinner
    and Lily is saint, this is false, consistent with Penelope being a sinner. Riley's statement "if
    Lily is sinner then Mia is sinner" has a false premise (Lily is saint), making it vacuously true,
    so Riley is a saint. Aria's statement "Lily is saint" is true, so Aria is a saint. <answer>
    Penelope is a sinner, Lily is a saint, Riley is a saint, Mia is a sinner, and Aria is a saint.</
    answer>

Example 9:
Question: A very special island is inhabited only by angels and devils. Angels always tell the truth,
    and devils always lie. You meet 6 inhabitants: Liam, Zoey, Ava, Logan, James, and Avery. Liam
    commented, "James is a devil if and only if Ava is a devil". Zoey asserted: "Zoey is an angel and
    Logan is a devil". In Ava's words: "Liam is a devil". "Avery is a devil," Logan mentioned. James
    said that if Avery is an angel then Zoey is an angel. Avery asserted: "Avery is an angel and Liam
    is a devil". So who is an angel and who is a devil? (Format your answer like: "Liam is a angel/
    devil, Zoey is a angel/devil, Ava is a angel/devil, Logan is a angel/devil, James is a angel/devil
    , and Avery is a angel/devil")
Answer: Assume Ava is an angel. Her statement "Liam is devil" means Liam is a devil. Liam's statement "
    James is devil <-> Ava is devil" - if Liam is a devil (lies), since Ava is angel (not devil), for
    the biconditional to be false, James must also not be a devil, so James is a devil (making both
    sides false, thus true). Wait, let me reconsider. If Liam lies, the biconditional is false. Ava is
    angel (not devil), so for false biconditional, James must be devil. Assume Avery is an angel.
    Avery's statement "Avery is angel AND Liam is devil" is true. Zoey's statement "Zoey is angel AND
    Logan is devil" - if this is false, Zoey is a devil. Logan's statement "Avery is devil" is false,
    so Logan is a devil. James's statement "if Avery is angel then Zoey is angel" - since Avery is
    angel and Zoey is devil, this is false, so James is a devil. <answer>Liam is a devil, Zoey is a
    devil, Ava is an angel, Logan is a devil, James is a devil, and Avery is an angel.</answer>

Example 10:
Question: A very special island is inhabited only by knights and knaves. Knights always tell the truth,
    and knaves always lie. You meet 6 inhabitants: Aria, Ava, Amelia, Grace, Charlotte, and Jack. "
    Jack is a knight," Aria claimed. In a statement by Ava: "Jack is a knight". Amelia asserted: "Jack
    is a knave and Grace is a knight". Grace commented, "Aria is a knight if and only if Charlotte is
    a knave". As Charlotte put it, "Aria is a knight". Jack noted, "Ava is a knave if and only if
    Charlotte is a knave". So who is a knight and who is a knave? (Format your answer like: "Aria is a
    knight/knave, Ava is a knight/knave, Amelia is a knight/knave, Grace is a knight/knave, Charlotte
    is a knight/knave, and Jack is a knight/knave")
Answer: Assume Aria is a knight. Her statement "Jack is knight" means Jack is a knight. Ava's statement
    "Jack is knight" is true, so Ava is a knight. Charlotte's statement "Aria is knight" is true, so
    Charlotte is a knight. Jack's statement "Ava is knave <-> Charlotte is knave" - both Ava and
    Charlotte are knights (not knaves), so both sides are false, making the biconditional true,
    consistent with Jack being a knight. Grace's statement "Aria is knight <-> Charlotte is knave" -
    Aria is knight and Charlotte is knight (not knave), so the biconditional is false, meaning Grace
    is a knave. Amelia's statement "Jack is knave AND Grace is knight" - Jack is knight and Grace is
    knave, so this is false, meaning Amelia is a knave. <answer>Aria is a knight, Ava is a knight,
    Amelia is a knave, Grace is a knave, Charlotte is a knight, and Jack is a knight.</answer>
```

# LLM USE

LLMs were used to revise and proofread paper content. All claims have been verified and cross-referenced by the authors.

