# OpenReview forum: "Learning from Synthetic Data Improves Multi-hop Reasoning"
_ICLR.cc/2026/Conference — ICLR 2026 Poster_

### Official Review · Reviewer_hyPz · 2025-10-23

**Soundness:** 3
**Presentation:** 4
**Contribution:** 4
**Rating:** 6
**Confidence:** 5

**Summary:**

This paper investigates an important and timely question: whether fine-tuning Large Language Models (LLMs) on synthetic data through reinforcement learning, in the absence of real-world knowledge, can improve their multi-hop reasoning capabilities on real-world question-answering (QA) tasks. The authors use two synthetic datasets: GSM-∞ (math word problems) and PhantomWiki (fictional knowledge-base QA), to fine-tune several open-source models of varying scales (Qwen series, Phi-4-mini-reasoning). The experimental results show that despite the synthetic data having no factual overlap with the real-world evaluation benchmarks (such as HotpotQA, 2WikiMultihopQA, and MuSiQue), the fine-tuned models achieve significant performance improvements on these real-world tasks. The authors argue that this improvement stems from the model learning a transferable meta-skill—"knowledge composition," the ability to chain multiple logical inference steps. The study also finds that this performance gain is consistent across different model families and sizes, and does not suffer from overfitting as training data increases.

**Strengths:**

1. As high-quality human-annotated data becomes increasingly scarce, exploring the value of synthetic data is a frontier direction in the LLM field. This paper explores the fundamental question of whether reasoning abilities can be learned independently of factual knowledge, which has significant implications for the training strategies of Large Language Models (LLMs).
2. The use of completely disjoint synthetic and real-world datasets effectively controls for memorization, providing clearer evidence for skill transfer.
3. The paper not only reports final results but also provides an in-depth analysis of the model's learning process by examining performance changes during training and stratifying performance by question difficulty. In particular, Figure 3 and Figure 5 clearly demonstrate how the model's progress on more difficult synthetic questions translates to performance improvements on real-world tasks.
4. The paper is well-structured and flows logically from introduction to conclusion. The research motivation, methodology, and results are all clearly articulated. The figures (especially Figure 2 and Figure 5) intuitively present the core findings and are easy to understand.

**Weaknesses:**

1. The training is conducted exclusively on synthetic data. Although this shows improvement in real-world scenarios, the paper lacks a comparison with a baseline trained on real-world data and tested on real-world data. There is no analysis of the potential performance gap between training on synthetic data versus real-world data.
2. The experimental models are relatively small, with the largest being 4B. It would be beneficial to see experiments on models of at least 7B parameters. Based on Figures 2 and 3, the performance improvement for Qwen3-1.7B is notably smaller than that for Qwen3-0.6B. Could this be because the 0.6B model has weaker baseline reasoning abilities, and thus training naturally improves its generalizable reasoning performance, while the larger 1.7B model shows diminished gains? What would happen with an even larger model, such as a Qwen3-8B? Would the improvement be minimal?
3. The paper states that 3 (for GSM-∞) or 11 (for PhantomWiki) CoT examples are used during RL training. Is it necessary to include these CoT examples in the prompt during inference as well? Furthermore, it would be desirable to see experiments on the training and generalization effects in a Zero-Shot setting, i.e., without including any CoT examples.
4. In Figure 2, the training results on GSM-∞ are consistently worse than on PhantomWiki for nearly all models. Is there a deeper explanation for this? Is it because GSM-∞ focuses more on mathematical reasoning rather than the type of multi-hop reasoning found in the evaluation tasks? If so, does this raise concerns about the generalizability of the reasoning skills being learned?

**Questions:**

See weaknesses

---

> ### Author Response · Authors · 2025-11-21
>
> We are glad that you find our contributions well-structured, clear, and accessible. We agree that using programmatically-generated synthetic datasets controls for memorization, isolates knowledge composition as a learnable skill, and enables a fine-grained analysis on model improvement. We address your questions below:
>
> > no analysis of the potential performance gap between training on synthetic data versus real-world data
>
> Good suggestion! We RL **fine-tuned all small models with HotpotQA, 2Wiki, and MuSiQue** training datasets. The training setup is exactly the same as with synthetic datasets (10K training samples, F1 scores as reward etc.). We do observe a performance gap as you hypothesize, see Figure 10 in appendix and https://imgur.com/a/R70PZ8m. RL fine-tuning on human-annotated data outperforms programmatically-generated synthetic datasets (compared to PhantomWiki results).
>
> We believe that this **performance gap is expected as _in-domain_ training data is usually better than _out-of-domain_**: real-world articles and question-answer pairs have rich semantic complexity and constraints, whereas synthetic data have limited linguistic style as they are generated from templates. On the contrary, such kind of synthetic data is free and scalable. We find that programmatically-generated synthetic data can bridge the gap to real-world data as a free data source.
>
> > experimental models are relatively small, with the largest being 4B
>
> > training naturally improves its [Qwen3-0.6B’s] generalizable reasoning performance, while the larger 1.7B model shows diminished gains? What would happen with an even larger model?
>
> Our findings on 1-4B models show emergent reasoning on small models, challenging widespread assumptions that larger models are necessary. We agree that understanding larger model behavior is important, hence we **extend experiments to GRPO full-fine-tuning Qwen2.5-7B-Instruct (7B parameters) and Qwen3-4B** (4B parameters, comparable to Phi-4-mini-reasoning), see Figure 6 in the appendix (https://imgur.com/a/UtmoyDY). More details in the general comment.
>
> Briefly, these larger models showed strong baseline performance, and RL fine-tuning on either synthetic or real-world data did not further improve evaluation scores. Due to our limited compute capacity, we launched these initial experiments on larger models with the absolute minimum train batch sizes and the number of GRPO rollouts. We are actively working to improve RL fine-tuning on larger models, but that might require access to compute unavailable in our academic lab setting.
>
> With tight yet extensive experimentation on 1-4B model sizes, our paper’s main contribution still stands: knowledge composition is a transferable skill that can be learned in purely fictional worlds.
>
> > Is it necessary to include these CoT examples in the prompt during inference
> > desirable to see experiments on the training and generalization effects in a Zero-Shot setting
>
> Much work establishes that CoT examples improve the base model’s performance on reasoning benchmarks over zeroshot prompting [https://arxiv.org/abs/2201.11903]. So by including CoT examples at inference, we compare base and trained models on equal grounds. Due to limited compute, we are not able to run a parallel set of experiments in a zeroshot setting.
>
> > training results on GSM-∞ are consistently worse than on PhantomWiki for nearly all models
>
> > GSM-∞ focuses more on mathematical reasoning rather than the type of multi-hop reasoning
>
> This is an interesting hypothesis. From our experiment results, it is difficult to argue “reasoning domain” as the key cause for optimal synthetic-to-real transfer. We did find a minor bug in GSM-Infinite reward calculation, and reran our training experiments—synthetic-to-real transfer improves slightly and is sometimes competitive with PhantomWiki.
>
> We hesitate to claim that one training dataset is always better, since one would need to control for various confounding factors such as reasoning domain, question difficulty levels, and linguistic diversity. **Our main contribution is in demonstrating the utility of programmatically-generated synthetic data**. Characterizing features of synthetic data that fully unlock real-world multi-hop reasoning capabilities is an exciting future direction.

---

### Official Review · Reviewer_F2Sn · 2025-10-26

**Soundness:** 2
**Presentation:** 2
**Contribution:** 2
**Rating:** 4
**Confidence:** 3

**Summary:**

The paper investigates whether RL fine-tuning on purely synthetic data constructed to require multi-hop reasoning but to contain no real-world facts can teach a general skill of knowledge composition that then transfers to established, real benchmarks. Concretely, the authors fine-tune small LLMs  with GRPO on two synthetic sources. Experiments show that RL on synthetic data improves performance on all three downstream datasets and across model families and sizes.

**Strengths:**

* I appreciate that the paper presents a tight experimental design around a single, interpretable hypothesis: i.e., training on universes that are explicitly non-overlapping with real-world knowledge, the study isolates whether multi-hop structure learned in synthetic settings can carry over.

* The paper also tackles an important topic in reasoning/RL, namely the effort to disentangle answer formatting from reasoning.

* I also like how the paper shows curves over checkpoints and stratifying performance by question difficulty (number of hops in PhantomWiki and number of operations in GSM infinite), which provides a richer picture than just showing single endpoint scores.

**Weaknesses:**

* Although the empirical story is neatly organized, in my view, the novelty is modest given the rapidly expanding literature on synthetic data and RL for reasoning.
  * If I remember correctly, PhantomWiki itself was introduced as an on-demand synthetic universe generator to test reasoning and retrieval while sidestepping data leakage; it feels more like this paper leverages that dataset rather than advancing the generation framework. Likewise, GSM-infinite was created to probe reasoning under controllable arithmetic complexity and long contexts, and here it serves as a training curriculum rather than as a novel contribution.
  * Similarly from the methodology side, the paper’s RL component employs GRPO but it seems like without much methodological innovation; recent works (a la Deepseek) have shown to a degree that RL alone can elicit sophisticated reasoning behaviors without human step-by-step traces and RLVR can better incentivize process-correctness. The present study feels like a combination and transfer evaluation rather than as a new algorithm etc. The finding of using synthetic data to help with reasoning doesn't seem particular novel to me either.

* If I understand correctly, one of the paper's claims is that knowledge composition is the specific causal skill that helps with performance; in terms of evidence, the paper infers this composition skill mainly from higher F1 values on 2-4 hop datasets and from difficulty-stratified curves; however, I don't see where the paper verifies whether the intermediate steps followed by the model are logically correct and path-faithful?

* While the focus is on RL, I don't see any other baselines (e.g., SFT) on the same synthetic datasets: how do those compare? Without some of these comparisons, it's unclear to me if gains are from RL itself or via additional supervised exposure to reasoning-style data (or something else). The models used seem to also all be sub 4B; I'm not sure how transferable these findings are in generality (e.g., to even 7/8B models or eve 3-4B modes outside of phi-mini) especially since results show that different models show different degrees "malleability" to RL (but the paper defers analysis to future work).

* I see that the evaluations are only on 3 datasets (HotpotQA, 2Wiki, MuSiQue); furthermore, results are done on sub-samples ($n=500$ with two seeds if I understand correctly; there is no re-sampling across multiple draws or paired testing. Combined with RL and small models (with potentially high variance), this seems quite small. I wonder how robust the gains shown here are to actual larger samples, increased repeated rates of sampling, etc.

**Questions:**

See weaknesses.

---

> ### Author Response · Authors · 2025-11-21
>
> Thank you for the feedback! We are delighted that you find our experiments well-controlled on testing a concrete hypothesis: whether knowledge composition is a transferable skill. We appreciate that our stratified visualizations on model improvement render a rich and fine-grained picture beyond improvements in final evaluation scores.
>
> > the novelty is modest given the rapidly expanding literature on synthetic data and RL for reasoning
>
> Recent works have shown transfer from LLM-generated synthetic data, but that from programmatically-generated synthetic datasets (questions and articles generated from templates) have not been fully established. Our **main contribution is in systematically illustrating transfer from programmatically-generated synthetic datasets to real-world multi-hop reasoning, which is novel to our knowledge**. See the general comment for details.
>
> As you remark, fictional-world datasets allow us to probe whether RL fine-tuning can develop the knowledge composition skill independently of real-world factual knowledge. We provide evidence for this hypothesis through **extended experiments across 4 synthetic training datasets, 5 evaluation benchmarks, and 6 LLMs from different model families** https://imgur.com/a/UtmoyDY. As we discuss in the general comment, the linguistic gap between such synthetic datasets and real-world multi-hop reasoning benchmarks is wide, and transfer does not follow immediately.
>
> While we do not introduce a new dataset or an RL algorithm, our experiments present a new perspective on the key elements necessary for real-world multi-hop reasoning. One such key element we identify is knowledge composition, which transfers and can be learned in fictional worlds.
>
> > I don't see where the paper verifies whether the intermediate steps followed by the model are logically correct
>
> In Figure 5 (previously Figure 4) we had plotted the frequency of model generations that contain correct intermediate answers of the MuSiQue benchmark. We found that, with synthetic data training, model generations contained higher fractions of ground-truth intermediate answers (orange lines moving upwards from base model). We have **added another multi-hop reasoning benchmark (CofCA) to the figure** https://imgur.com/a/Ib4x669. See the general comment for more details.
>
> > I don't see any other baselines (e.g., SFT)
>
> > gains are from RL itself or via additional supervised exposure to reasoning-style data
>
> SFT requires ground-truth solution text or reasoning traces for distillation. Programmatically-generated synthetic datasets like PhantomWiki and ReasoningGym do not include solution text, and collecting reasoning traces from frontier LLMs can be prohibitively expensive.
>
> GSM-Infinite is a special case where the dataset generates ground-truth solution text from templates. We **found that SFT on GSM-Infinite solution texts can overfit to the arithmetic synthetic task and show limited transfer to real-world multi-hop reasoning**. In contrast, RL fine-tuning developed the generalizable knowledge composition skill. See details in the general comment.
>
> > transferable these findings are in generality (e.g., to even 7/8B models or eve 3-4B modes outside of phi-mini) especially since results show that different models show different degrees "malleability" to RL
>
> Our findings on 1-4B models show emergent reasoning on small models, challenging widespread assumptions that larger models are necessary. We agree that understanding larger model behavior is important, hence we **extend experiments to GRPO full-fine-tuning Qwen2.5-7B-Instruct (7B parameters) and Qwen3-4B** (4B parameters, comparable to Phi-4-mini-reasoning), see Figure 6 in the appendix (https://imgur.com/a/UtmoyDY). More details in the general comment.
>
> Briefly, these larger models showed strong baseline performance, and RL fine-tuning on either synthetic or real-world data did not further improve evaluation scores. Due to our limited compute capacity, we launched these initial experiments on larger models with the absolute minimum train batch sizes and the number of GRPO rollouts. We are actively working to improve RL fine-tuning on larger models, but that might require access to significantly more compute unavailable in our academic lab setting. It is premature to comment on the RL-ability of the larger models just from these experiments, and we aim to understand it better with future work.
>
> With tight yet extensive experimentation on 1-4B model sizes, our paper’s main contribution still stands: knowledge composition is a transferable skill that can be learned in purely fictional worlds.

---

> > ### Author Response · Authors · 2025-11-21
> >
> > > results are done on sub-samples ( with two seeds if I understand correctly
> >
> > > evaluations are only on 3 datasets (HotpotQA, 2Wiki, MuSiQue)
> >
> > > Combined with RL and small models (with potentially high variance)
> >
> > > how robust the gains shown here are
> >
> > We had RL fine-tuned every (training dataset, model) combination with 2 independent training random seeds, as mentioned in Figure 2 caption. All evaluation numbers were averaged across the 2 training random seeds and on a random subsample of evaluation benchmarks. This ensured noise due to RL was controlled for. We also reported standard errors in our evaluation measurements, and showed significance with confidence intervals.
> >
> > We have **extended our experiments to 2 new evaluations benchmarks, 2 new synthetic data generators, and 2 larger models**. See more details in the general comment.

---

### Official Review · Reviewer_3JJe · 2025-10-30

**Soundness:** 3
**Presentation:** 3
**Contribution:** 2
**Rating:** 4
**Confidence:** 3

**Summary:**

This paper examines whether large language models can acquire general reasoning abilities solely from synthetic data, without relying on real-world knowledge.
Using reinforcement learning on fully artificial datasets such as PhantomWiki and GSM, the authors show significant performance improvements on real-world multi-hop QA benchmarks.
The results suggest that reasoning skills, such as knowledge composition, can transfer across domains, highlighting synthetic data as a scalable alternative to human-annotated reasoning datasets.

**Strengths:**

The study demonstrates that large language models can acquire generalizable reasoning skills purely from synthetic, knowledge-free data. It provides empirical evidence that these synthetic reasoning abilities transfer to real-world multi-hop QA tasks, achieving substantial performance gains. The approach offers a scalable and cost-effective framework for improving reasoning through verifiable, automatically generated training data.

**Weaknesses:**

- The experiments are limited to multi-hop QA. Even though the training data come from a synthetic world, the fact that performance improves on other multi-hop QA benchmarks is not particularly surprising.
- The applicability of the approach to grammatically or semantically complex real-world texts remains unknown.

**Questions:**

None

---

> ### Author Response · Authors · 2025-11-21
>
> We are pleased that you find our work presenting **clear empirical evidence** of synthetic-to-real transfer in multi-hop reasoning. Indeed, models can acquire generalizable reasoning skills from synthetic fictional datasets, offering a scalable and inexpensive data resource for improving reasoning.
>
> > The experiments are limited to multi-hop QA. Even though the training data come from a synthetic world, the fact that performance improves on other multi-hop QA benchmarks is not particularly surprising.
>
> > The applicability of the approach to grammatically or semantically complex real-world texts remains unknown.
>
> We clarify the significance of our findings. Our programmatically-generated synthetic training datasets (PhantomWiki, GSM-Infinite, ReasoningGym) use rule-based templates and context-free-grammars to generate questions and articles. This not only removes factual knowledge useful in the real-world, but also limits their linguistic style and semantic complexity. In contrast, all 5 multi-hop evaluation benchmarks are curated from real-world Wikipedia text, and require resolving hops and constraints on semantically complex questions.
>
> **Due to the linguistic gap between programmatically-generated synthetic datasets and real-world multi-hop reasoning benchmarks, the transfer is not obvious**. In the general comment, we emphasize the significance of our synthetic-to-real transfer contributions.
>
> To further substantiate our claims, we have added **2 new evaluation benchmarks (CofCA and SynthWorlds-RM)** that disentangle memorized knowledge from knowledge composition skill in frontier LLM performance. RL fine-tuning on synthetic datasets transfers to these semantically complex benchmarks as well (updated Figure 2 https://imgur.com/a/UtmoyDY). In Figure 5, we plotted intermediate answer analysis on MuSiQue and CofCA benchmarks https://imgur.com/a/Ib4x669 to **visualize that, as synthetic training progresses, models generations contain higher fractions of correct intermediate steps**. See general comment for more details.
>
> Refs:
> - PhantomWiki: https://arxiv.org/abs/2502.20377
> - HotpotQA: https://arxiv.org/abs/1809.09600
> - Counterfactual QA (CofCA): https://arxiv.org/abs/2402.11924v5
> - SynthWorlds: https://arxiv.org/abs/2510.24427

---

### Official Review · Reviewer_hJzK · 2025-11-01

**Soundness:** 3
**Presentation:** 2
**Contribution:** 2
**Rating:** 4
**Confidence:** 4

**Summary:**

The paper shows that RL fine-tuning (GRPO) on purely synthetic multi-hop datasets (GSM-infinity, PhantomWiki) improves LLM performance on real-world QA benchmarks (HotpotQA, 2Wiki, MuSiQue) by teaching a transferable “knowledge composition”.

**Strengths:**

- Provides clear empirical evidence that RL fine-tuning on synthetic datasets (PhantomWiki, GSM-infinity) improves LLM multi-hop reasoning on real-world QA benchmarks.

- Addresses a practical problem, the scarcity and cost of high-quality human annotated, which the paper suggests can be supplemented or replaced by synthetic reasoning data.

- Demonstrates consistent performance gains across multiple model families and parameter scales, indicating robustness and generalizability.

- The experimental details and use of open-source technologies (models and codebase) makes the setup reproducible.

**Weaknesses:**

- The paper’s novelty is limited as prior works have already shown that synthetic data and SFT/RLVR for reasoning works quite well. The contribution is primarily about a different reasoning setup of multi-hopping.

- The domain of synthetic data is narrow, focusing only on arithmetic and relational reasoning, which limits claims of general reasoning transfer.

- The evaluation datasets lack diversity. HotpotQA, 2WikiMultihopQA, and MuSiQue are all two-hop or near two-hop QA tasks, reducing the strength of the generalization claim.

- The paper is unnecessarily verbose at places. The related work section covers too much ground that's not relevant. Stating GRPO equations in Section 3.2 was not really necessary.

Minor nit:
- reinforcement -> Reinforcement
- L128: “complicated, RL-based framework” -> why complicated?

**Questions:**

-  In the abstract, there's this phrase "“high scoring latency” which is not clear to me. Can you please explain?

- Is there any reason to prefer RL over SFT in this setup?

---

> ### Author Response · Authors · 2025-11-21
>
> Thank you for your feedback. We are glad that you find our contributions **clear, practical, and consistent** with well-controlled experiments. We address all comments below.
>
>
> > paper’s novelty is limited as prior works have already shown that synthetic data and SFT/RLVR for reasoning works quite well
>
> We study RL fine-tuning on programmatically-generated synthetic data that use templates for generating questions, as opposed to LLM-generated synthetic data. In the general comment, we discuss why our work stands out from recent work on SFT/RLVR with synthetic data. Notably, our contributions demonstrate that **knowledge composition skill can be learned in fictional worlds and transferred to real-world use**. This has significant implications in fine-tuning LLMs with inexpensive and error-free synthetic datasets to enhance reasoning capabilities.
>
> > domain of synthetic data is narrow, focusing only on arithmetic and relational reasoning
> > The evaluation datasets lack diversity
>
> This is a valid concern, and we have **added two more synthetic data generators from ReasoningGym**: "knights-knaves (logic puzzle on truth tables) and "family-relationships (another relational reasoning). Moreover, we have added two recent multi-hop reasoning benchmarks with higher complexity (CofCA and SynthWorlds-RM). We continue to observe synthetic-to-real transfer. See general comment for details, and updated Figure 2 https://imgur.com/a/UtmoyDY.
>
> > The paper is unnecessarily verbose at places
>
> Thank you for pointing this out. We have revised all sections for accessibility and succinctness. We moved GRPO technical details to the appendix to make space for newer experiments. We will clarify our contributions in the context of related work.
>
> > L128: “complicated, RL-based framework” -> why complicated?
> > reinforcement -> Reinforcement
>
> Complicated technically over CoT prompting and next-token prediction in SFT, which are also used for model alignment. RLHF requires scoring and collecting chosen/rejected text pairs, potentially training reward models, and hyperparameter tuning to stabilize training. We will add your suggestions.
>
> > In the abstract, there's this phrase "“high scoring latency” which is not clear to me. Can you please explain?
>
> GRPO training requires scoring model generations on-the-fly as the RL algorithm is on-policy. LLM-in-the-loop verifiers inherently add latency to the scoring step. In contrast, programmatically-generated synthetic datasets with precomputed answers enable instant verification.
>
>
> > Is there any reason to prefer RL over SFT in this setup?
>
> We use programmatically-generated synthetic data for training, and do not rely on reasoning traces from other LLMs for distillation (see the general comment for the comparison). Programmatically-generated synthetic data is often constructed without ground-truth solution text, so obtaining verifiable traces for SFT is difficult.
>
>
> With that said, we have added an SFT vs RL experiment on GSM-Infinite—a dataset that uniquely does include ground-truth solution text. We found that SFT was more likely to overfit the model to the synthetic task, and had worse transfer to real-world multi-hop reasoning compared to RL fine-tuning. See general comment for more details.

---

### Author Response · Authors · 2025-11-21

We thank all reviewers for their constructive feedback. We appreciate that **all reviewers find our contributions well-motivated, practical, and clear**. They further note that our **work shows this synthetic-to-real transfer with well-controlled yet comprehensive experiments**, and **visualizes how the knowledge composition skill evolves during training**.

In response to the feedback, we have (1) clarified the novel contributions of our work (`hJzK`, `3JJe`, `F2Sn`), (2) expanded the evaluation scope (`hJzK`, `3JJe`, `F2Sn`), (3) included more diverse training datasets with (`hJzK`, `F2Sn`, `hyPz`), (4) added experiments with larger models (`F2Sn`, `hyPz`), and (5) added a comparison between SFT and RL fine-tuning strategies on synthetic datasets (`hJzK`, `F2Sn`).

## Clarification on novel contributions of our work

### LLM- vs programmatically-generated synthetic data

Our work focuses on **programmatically-generated synthetic data**, which we contrast with LLM-generated synthetic datasets. We summarize the differences between these two kinds of synthetic data below:
- Programmatically-generated data relies on strict templates and other tools like context-free grammars and logic programs. This makes it **semantically simpler but more verifiable** than LLM-generated synthetic data, which has richer linguistic style but is also harder to verify.
- Programmatically-generated data enables **precise control over question difficulty levels** and their structure. This allows for finer-grained analysis of both reasoning capabilities and fine-tuning strategies.
- Programmatically-generated data is often **cheaper to generate at scale**, as it does not require   querying a frontier LLM. This is valuable for developing smaller models under resource constraints..

While transfer benefits from LLM-generated synthetic data are well-established, transfer from programmatically-generated synthetic data to real-world use is under-explored [ReasoningGym: https://arxiv.org/abs/2505.24760].

### Investigating transfer capabilities of _factual knowledge-free_ synthetic data

**Second, we argue that synthetic-to-real transfer is not obvious**. Since programmatically-generated synthetic datasets use templates on fictional worlds, they contain no factual or semantic overlap with real-world multi-hop reasoning benchmarks like HotpotQA, 2Wiki, and MuSiQue. Grammar and semantic complexities in real-world multi-hop reasoning benchmarks go far beyond simple relationships in programmatically-generated synthetic data. See examples below.

Hence, we study a fundamental question: _does learning to compose knowledge in worlds lacking linguistic diversity transfer to realistic multi-hop tasks rich with semantic complexity?_ Our **contribution illustrates that LLMs can transfer the generalizable knowledge composition skill despite the linguistic complexity gap**. We demonstrate this with extended experiments on 4 synthetic training datasets, 5 evaluation benchmarks, and 6 LLMs from different model families, with fine-grained analysis on reasoning evolution during RL fine-tuning.

We will update our introduction to clarify our contributions in the context of related work.

- From PhantomWiki [Figure 2, https://arxiv.org/abs/2502.20377]: _“Who is the nephew of the friend of the person whose hobby is birdwatching?”]_
- From HotpotQA [Table 3, https://arxiv.org/abs/1809.09600]: _“Aside from Yodobashi, what other towns were merged into the ward which gave the major Japanese retail chain specializing in electronics, PCs, cameras, and photographic equipment it’s name?”_

---

> ### Author Response · Authors · 2025-11-21
>
> ## Addressing reviewer feedback
>
> ### Evaluation breadth (reviewers `hJzK`, `3JJe`, `F2Sn`)
>
> We have **added 2 recent multi-hop reasoning benchmarks with higher complexity: Counterfactual QA** (CofCA) and **SynthWorlds-RM**. These complement our 3 existing benchmarks (HotpotQA, 2Wiki, MuSiQue) and evaluate models on multi-hop questions spanning 2-6 hops and constraints. CofCA https://arxiv.org/abs/2402.11924v5 uses counterfactual knowledge augmentation to focus on more genuine reasoning, while SynthWorlds-RM https://arxiv.org/abs/2510.24427 leverages six distinct graph motifs to generate structurally complex reasoning chains.
>
> Results show consistent synthetic-to-real transfer across all benchmarks in all training setups (updated Figure 2 and new Figure 6 in appendix https://imgur.com/a/UtmoyDY). We have also **extended Figure 5 to visualize intermediate answer analysis on CofCA questions in addition to MuSiQue** https://imgur.com/a/Ib4x669. Similar to results on MuSiQue, LLM generations for CofCA contain larger fractions of correct intermediate answers as RL fine-tuning progresses on PhantomWiki synthetic data. This demonstrates the utility of synthetic data beyond final F1 scores.
>
> ### Training breadth (reviewers `hJzK`, `F2Sn`, `hyPz`)
>
> We have **added 2 new synthetic data generators from ReasoningGym** https://arxiv.org/abs/2505.24760: “family-relationships” (RG-Family) and “knights-knaves” (RG-Knights), see updated Figure 2 https://imgur.com/a/UtmoyDY. RG-Family requires inferring relationships and RG-Knights involves solving logic puzzles—both distinct from PhantomWiki (composing implications) and GSM-Infinite (arithmetic reasoning).
>
> We observe considerable synthetic-to-real transfer from both ReasoningGym datasets. However, PhantomWiki and GSM-Infinite achieve stronger transfer, making them better suited for in-context multi-hop reasoning. Identifying characteristics of programmatically-generated synthetic data that transfer best remains an important direction for future work.
>
> ### Model scale (reviewers `F2Sn`, `hyPz`)
>
> Given the common belief that larger models are “necessary” for reasoning, our extensive findings on 1-4B models demonstrate emergent reasoning on every scale. We present sweeping ablations across training datasets and evaluation benchmarks, and **paint a thorough and rich picture of _how_ RL fine-tuning can develop reasoning abilities on cheaply available synthetic data**. This has important implications in the community’s understanding of LLM reasoning.
>
> That said, extending experiments to larger models is important. We **extend experiments to full-fine-tuning larger models in the Qwen family with GRPO**: Qwen2.5-7B-Instruct (7B parameters) and Qwen3-4B (4B parameters, comparable to Phi-4-mini-reasoning), see Figure 6 in the appendix (https://imgur.com/a/UtmoyDY). On these larger models with strong baseline performance, we observe mixed results of RL fine-tuning on synthetic datasets; the synthetic-to-real transfer is limited. Moreover, we do not observe multi-hop reasoning improvement  **even when RL fine-tuning these larger models on real-world multi-hop reasoning data (HotpotQA).** Note that for each evaluation benchmark, standard errors of different models overlap.
>
> Qwen3-4B:
>
> |             | HotpotQA      | 2Wiki         | MuSiQue       | CofCA         | SynthWorlds-RM   |
> |-------------|---------------|---------------|---------------|---------------|------------------|
> | base        | 0.707 ± 0.018 | 0.781 ± 0.017 | 0.479 ± 0.021 | 0.608 ± 0.019 | 0.613 ± 0.020    |
> | PhantomWiki | 0.659 ± 0.018 | 0.734 ± 0.019 | 0.418 ± 0.021 | 0.616 ± 0.019 | 0.569 ± 0.020    |
> | HotpotQA    | 0.730 ± 0.017 | 0.782 ± 0.017 | 0.500 ± 0.020 | 0.665 ± 0.018 | 0.624 ± 0.020    |
>
> Qwen2.5-7B-Instruct:
>
> |             | HotpotQA      | 2Wiki         | MuSiQue       | CofCA         | SynthWorlds-RM   |
> |-------------|---------------|---------------|---------------|---------------|------------------|
> | base        | 0.659 ± 0.019 | 0.654 ± 0.020 | 0.377 ± 0.020 | 0.582 ± 0.019 | 0.560 ± 0.020    |
> | PhantomWiki | 0.692 ± 0.017 | 0.781 ± 0.017 | 0.461 ± 0.020 | 0.586 ± 0.019 | 0.525 ± 0.019    |
> | HotpotQA    | 0.736 ± 0.017 | 0.612 ± 0.021 | 0.458 ± 0.020 | 0.529 ± 0.020 | 0.621 ± 0.020    |
>
> Due to our limited compute in an academic setting, we launched these initial experiments on larger models with the absolute minimum train batch sizes and the number of GRPO rollouts. When training original small models, large batch sizes and number of rollouts had stabilized training. We are actively working to improve RL fine-tuning on larger models and getting access to more compute.
>
> With tight yet extensive experimentation on 1-4B model sizes, our paper’s main contribution still stands: knowledge composition is a transferable skill that can be learned in purely fictional worlds.

---

> > ### Author Response · Authors · 2025-11-21
> >
> > ### Supervised fine-tuning (SFT) vs RL fine-tuning (reviewers `hJzK`, `F2Sn`)
> >
> > We focus on **LLM fine-tuning when the ground-truth solution traces or reasoning traces from other LLMs (for distillation) are unavailable**. SFT requires such text, which is not generally available and expensive to collect from frontier LLMs. In contrast, RL only requires verifiable answers, which programmatically-generated synthetic datasets readily include. We investigate RL fine-tuning as a more general approach in this setting.
> >
> > While not our main focus, we compared SFT vs RL fine-tuning on GSM-Infinite—a special case of programmatically-generated synthetic data that includes ground-truth solution traces. We SFT Qwen3-0.6B and Qwen3-1.7B models on the training dataset’s solution traces. While this improved performance on a held-out GSM-Infinite dataset, we observed _no_ transfer benefits to real-world multi-hop reasoning benchmarks. This indicates that **SFT can overfit to the synthetic data task** while **RL fine-tuning yields better synthetic-to-real transfer.**
> >
> > _Accuracy on GSM-Infinite held-out set (in-domain)_
> >
> > | model      | base   | SFT     | RL    |
> > |------------|--------|--------|--------|
> > | Qwen3-0.6B | 0.0241 | 0.7735 | 0.6452 |
> > | Qwen3-1.7B | 0.1354 | 0.7742 | 0.8532 |
> >
> > _F1 scores on HotpotQA benchmark (transfer)_
> >
> > | model      | base   | SFT     | RL    |
> > |------------|--------|--------|--------|
> > | Qwen3-0.6B | 0.3569 | 0.3995 | 0.4786 |
> > | Qwen3-1.7B | 0.5935 | 0.5761 | 0.6664 |
> >
> > We sincerely thank all reviewers for constructive comments, which helped us strengthen our contributions. We will update our results to include the additional training seeds of these experiments as they become available.

---

### Meta-Review · Area_Chair_uTST · 2025-12-21

**Summary:**

_this contains verbatim comments from reviewers_
The paper aims to show that the LLMs can learn solely from synthetic data. Yet, the experiments are limited to multi-hop QA. Even though the training data come from a synthetic world, the fact that performance improves on other multi-hop QA benchmarks is not particularly surprising. The applicability of the approach to grammatically or semantically complex real-world texts remains unknown.

**Reviewer Concerns:**

_Reviewer 3JJe_
- The experiments are limited to multi-hop QA. Even though the training data come from a synthetic world, the fact that performance improves on other multi-hop QA benchmarks is not particularly surprising.
- The applicability of the approach to grammatically or semantically complex real-world texts remains unknown.

**Reviewer Scores:**

I don't think the scores could have changed with a better interaction with the reviewers.

---

### Decision · Program_Chairs · 2026-01-26

Accept (Poster)